# Energy-oriented Diffusion Bridge for Image Restoration with Foundational Diffusion Models

**Jinhui Hou[1], Zhiyu Zhu[1,2,*], and Junhui Hou[1,*]**
[1]Department of Computer Science, City University of Hong Kong;
[2]Department of Computer Science, City University of Hong Kong (Dongguan).
`jhhou3-c@my.cityu.edu.hk, zhiyu.zhu@cityu-dg.edu.cn,`
`jh.hou@cityu.edu.hk`

## Abstract

Diffusion bridge models have shown great promise in image restoration by explicitly connecting clean and degraded image distributions. However, they often rely on complex and high-cost trajectories, which limit both sampling efficiency and final restoration quality. To address this, we propose an Energy-oriented diffusion Bridge (E-Bridge) framework to approximate a set of low-cost manifold geodesic trajectories to boost the performance of the proposed method. We achieve this by designing a novel bridge process that evolves over a shorter time horizon and makes the reverse process start from an entropy-regularized point that mixes the degraded image and Gaussian noise, which theoretically reduces the required trajectory energy. To solve this process efficiently, we draw inspiration from consistency models to learn a single-step mapping function, optimized via a continuous-time consistency objective tailored for our trajectory, so as to analytically map any state on the trajectory to the target image. Notably, the trajectory length in our framework becomes a tunable task-adaptive knob, allowing the model to adaptively balance information preservation against generative power for tasks of varying degradation, such as denoising versus super-resolution. Extensive experiments demonstrate that our E-Bridge achieves state-of-the-art performance across various image restoration tasks while enabling high-quality recovery with a single or fewer sampling steps. Our project page is https://jinnh.github.io/E-Bridge/.

## 1 Introduction

Diffusion models (Ho et al., 2020; Song et al., 2021) have established a new state-of-the-art in generative modeling, with a profound impact on image restoration (IR). Two dominant paradigms have emerged. The first involves guiding a standard generative process, which starts from pure Gaussian noise, using the degraded image as a condition (Dhariwal & Nichol, 2021; Saharia et al., 2022). Although versatile, this approach suffers from slow inference speeds, as it must traverse a high-energy trajectory across a vast entropy gap from a noise distribution to the image manifold.

To create a more direct transformation, bridge diffusion models (Li et al., 2023; Liu et al., 2023; Yue et al., 2024; Zhu et al., 2025a; Wang et al., 2025b) have gained traction by constructing a stochastic process from the degraded image distribution to the clean one. By starting closer to the target, these models significantly reduce the number of sampling steps. However, the diffusion trajectories in these models are often predefined and sub-optimal, which is typically a fixed or polynomial interpolation. Such trajectories may not represent the most efficient path, or *geodesic*, on the complex data manifold. They often enforce a redundant "re-noising" phase before denoising, leading to unnecessarily high trajectory energy and limiting both restoration quality and computational efficiency. Although recent work has focused on leveraging powerful pretrained generative priors, they often rely on complex "stitching" mechanisms (Wang et al., 2025a) to connect two distinct diffusion processes, which can introduce inconsistencies.

---

*Corresponding author.

Instead of patching existing models, we argue that the optimal restoration trajectory should be a low-energy geodesic on the data manifold. To this end, we propose an Energy-oriented diffusion Bridge (E-Bridge) framework that fundamentally re-engineers the restoration process. We first construct a more efficient, low-energy trajectory by designing a novel bridge process that evolves over a shorter, more controllable time horizon. This process starts the reverse (restoration) trajectory from an entropy-regularized point, which is a mixture of the degraded image and Gaussian noise, thus theoretically reducing the required control energy and bypassing the inefficient re-noising phase.

To ensure this low-energy trajectory approximates a geodesic, we innovatively leverage a pretrained denoiser not as a separate component, but as an integrated dynamic geodesic guidance field, continuously pulling the trajectory toward the natural image manifold. To traverse this trajectory with maximum efficiency, we draw inspiration from consistency models (Song et al., 2023; Lu & Song, 2025). We learn a single-step mapping function, optimized via a continuous-time consistency objective tailored for our trajectory, which can analytically map any state on the trajectory directly to the final restored image. This eliminates the need for slow, iterative sampling. Notably, the trajectory length in our framework becomes a tunable, task-adaptive knob. This allows our E-Bridge to adaptively balance information preservation (for mild degradations like denoising) against generative power (for severe degradations like super-resolution). Extensive experiments demonstrate that our E-Bridge achieves state-of-the-art performance across a variety of image restoration tasks, while enabling high-quality recovery in a single or very few sampling steps.

In summary, the main contributions of this work are:

- we propose a novel energy-oriented diffusion bridge framework with low-energy data manifold geodesic trajectories and adaptation to current large pretrained models, and give corresponding theoretical analysis compared with state-of-the-art methods;
- we give a closed-form consistency solver that circumvents the need for numerical iteration, enabling direct and single-step inference along the geodesic trajectory.

## 2 RELATED WORK

**Image Restoration via Bridge Models.** Recent advances in diffusion-based restoration have moved beyond simple conditioning towards bridge models (Bortoli et al., 2021; Luo et al., 2023; Zhou et al., 2024; Shi et al., 2023) that define a stochastic trajectory from degraded to clean images. Early works formulated this trajectory as a standard Brownian Bridge (BBDM) (Li et al., 2023; Liu et al., 2023) or as an optimal transport problem solved via Schrödinger Bridges (SB) (Wang et al., 2025b). Frameworks like UniDB (Zhu et al., 2025a) further unify these concepts. However, a common thread unites these models: they learn a score or drift function that requires a slow, iterative sampling process for inference. To overcome the inference bottleneck, UniDB++ (Pan et al., 2025) proposes a post-hoc acceleration, where a multi-step model is compressed into a single-step one. The Implicit I2SB (I3SB) (Wang et al., 2025b) introduces a non-Markovian sampling scheme for pre-trained I2SB models.

**Image Restoration using Conditional Diffusion Models.** Another major category of methods conditions a standard generative process that begins from pure noise. Early works (Saharia et al., 2022; Lugmayr et al., 2022; Hou et al., 2023; Zhang et al., 2023) train a conditional model from scratch, generating a high-quality image guided by the degraded input (Saharia et al., 2022). A different strategy involves guiding a pre-trained unconditional model during the inference stage. Notable examples include DDRM (Kawar et al., 2022), which leverages spectral projections to solve various linear inverse problems, and DiffPIR (Zhu et al., 2023), which employs Plug-and-Play optimization to incorporate degradation-specific priors. Recently, the focus has shifted towards efficiently adapting large-scale pre-trained models. This is typically accomplished by fine-tuning task-specific modules, such as the time-aware encoder in StableSR (Wang et al., 2024) or the multi-task, prompt-based adapter in UniRestore (Chen et al., 2025).

## 3 PRELIMINARIES

The task of image restoration can be conceptualized as transporting a degraded image $\mathbf{Y}$, residing on a low-quality manifold $\mathcal{M}_L$, to its corresponding clean counterpart $\mathbf{X}_0$ on a high-quality manifold $\mathcal{M}_H$. Existing diffusion models approach this via two primary ways:

**Guided Generative Trajectories.** These methods (Dhariwal & Nichol, 2021; Saharia et al., 2022; Zhu et al., 2025b) guide a standard reverse diffusion process, which starts from pure Gaussian noise ($\mathbf{X}_T \sim \mathcal{N}(0, \mathbf{I})$), towards the clean image $\mathbf{X}_0$ using the degraded image $\mathbf{Y}$ as a condition. From a Stochastic Optimal Control (SOC) (Kappen, 2008; Berner et al., 2022; Park et al., 2024; Zhu et al., 2025a) perspective, the efficiency of this trajectory is measured by its *energy*, a cost functional representing the total transportation between distributions:

$$J(\mathbf{u}) = \mathbb{E} \int_0^T \frac{1}{2} \left( \underbrace{\|\dot{\mu}(t)\|^2}_{(A)} + \underbrace{\|\dot{\mu}(t) - \mathbb{E}[b_{total}(t, \mathbf{X}_t)]\|^2}_{(B)} \right) dt, \quad (1)$$

where $\dot{\mu}(t)$ denotes the instantaneous velocity vector of the mean curve, while $\mathbb{E}[b_{total}]$ represents the effective drift field provided by the network. The term (A) defines the transportation energy, quantifying the kinetic cost associated with the velocity profile along the trajectory of $\dot{\mu}(t)$. In contrast, term (B) corresponds to the control energy, measuring the cost required to steer the dynamics along the path, which can also be parameterized to $\gamma \|\mathbb{E}\mathbf{X}_T - \mathbf{Y}\|^2$. Consequently, the optimal scenario is to achieve transport via a low-energy geodesic trajectory that requires minimal control intervention.

Advanced methods like IRBridge (Wang et al., 2025a) attempt to mitigate this by leveraging powerful pre-trained generative models. However, this introduces its own complexities, such as the *state distribution mismatch* between the pre-trained model's process and the desired restoration process. To bridge this gap, IRBridge proposes a transition equation that mathematically maps a state $\mathbf{X}_i^\circ$ from the restorative trajectory to a state $\mathbf{X}_j^*$ on the generative trajectory, using an estimate of the clean image $\hat{\mathbf{X}}_0$:

$$\mathbf{X}_j^* = \alpha \cdot \mathbf{X}_i^\circ + \beta \cdot \hat{\mathbf{X}}_0 + \gamma + \sigma \boldsymbol{\epsilon}. \quad (2)$$

However, this approach involves the complexity of coupling two distinct and separately defined processes, creating an indirect and potentially fragile connection between the degraded and clean image domains.

**Direct Bridge trajectories.** These approaches (Liu et al., 2023; Zhou et al., 2024; Yue et al., 2024) construct a more direct stochastic process from $\mathbf{Y}$ to $\mathbf{X}_0$. The simplest form is the standard Brownian Bridge, defined by the stochastic differential equation (SDE) that pins a Wiener process to start at $\mathbf{X}_0$ and end at $\mathbf{Y}$. In practice, rather than directly solving this SDE, models like I2SB (Liu et al., 2023) are built upon an analytical posterior distribution $p(\mathbf{x}_t|\mathbf{X}_0, \mathbf{Y})$. For example, I2SB defines the state $\mathbf{X}_t$ as:

$$\mathbf{X}_t \sim \mathcal{N}\left(\alpha_t \mathbf{X}_0 + (1 - \alpha_t)\mathbf{Y}, \sigma_t^2 \mathbf{I}\right), \quad (3)$$

where the variance $\sigma_t^2$ is often a symmetric, arch-shaped function (e.g., $\sigma_t^2 \propto t(1-t)$). The symmetric noise schedule generally forces the reverse process to undergo a "re-noising" phase before denoising. For an asymmetric task whose goal is unidirectional denoising and restoration, this initial noise addition process is redundant in terms of energy, increasing the random process energy of the path. In pursuit of a theoretically optimal path, other works explore Schrödinger Bridges, which reframe the problem as finding a trajectory measure $\mathcal{P}$ that minimizes the Kullback-Leibler (KL) divergence from a reference Wiener process measure $\mathcal{W}$, subject to the marginal constraints:

$$\min_{\mathcal{P}} \mathrm{KL}(\mathcal{P}\|\mathcal{W}) \quad \text{s.t.} \quad \mathcal{P}_0 = p(\mathbf{X}_0), \mathcal{P}_T = p(\mathbf{Y}). \quad (4)$$

Although the formulation in Eq. (4) aims to find the lowest-energy path, its standard solver, the Iterative Proportional Fitting (IPF) algorithm, is computationally prohibitive and often unstable for high-dimensional data, rendering it impractical. A more fundamental limitation, common to all aforementioned paradigms, is their reliance on iterative, multi-step sampling to approximate the solution of the underlying differential equation. This process, $\mathbf{x}_{t-\Delta t} \approx \mathbf{x}_t - \mathbf{v}(\mathbf{x}_t, t)\Delta t$, has a computational complexity of $\mathcal{O}(N \times C)$, where $N$ is the number of sampling steps. This inherent sequential dependency fundamentally restricts inference speed and prevents real-time applications.

## 4 PROPOSED METHOD

As previously analyzed, existing diffusion-based image restoration bridges are often encumbered by complex, computationally expensive, iterative sampling along high-energy restoration trajectories. To overcome these fundamental limitations, we introduce the Energy-oriented diffusion Bridge (E-Bridge), a novel framework that re-engineers the restoration process for unparalleled efficiency and

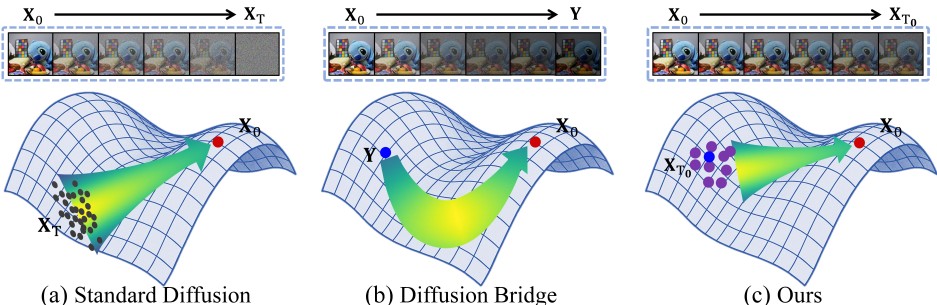

| (a) Standard Diffusion | (b) Diffusion Bridge | (c) Ours |

Figure 1: Illustration of diffusion processes for image restoration. (a) Standard Diffusion Models: These traverse a long, high-energy trajectory starting from pure Gaussian noise to the clean image manifold, conditioned on the degraded image. (b) Conventional Bridge Models: These construct a path from the degraded to the clean image but often follow a sub-optimal, high-energy trajectory that includes a redundant "re-noising" phase before denoising. (c) Our E-Bridge: It starts the reverse process from an entropy-regularized point, which is a mixture of the degraded image and noise, thus bypassing the inefficient re-noising phase and creating a more direct and shorter path for restoration.

fidelity. Our approach is built on a dual-pronged strategy. To be specific, first, we introduce the Energy-oriented Diffusion Bridge (E-Bridge), a novel, low-energy, data manifold geodesic trajectory. This trajectory is theoretically more efficient and better suited to the unidirectional nature of image restoration, avoiding the unnecessary re-noising phases of previous bridge models (Sec. 4.1). Second, we derive the E-Bridge-Solver, a principled, single-step mapping function obtained through the analytical inversion of our trajectory equation. Crucially, based on the E-Bridge-Solver, we train the E-Bridge by enforcing a continuous-time consistency optimization objective to further boost its performance. This ensures the solver produces a stable, high-fidelity output from any point on the trajectory, effectively learning the underlying geometry of the data manifold (Sec. 4.2).

## 4.1 ENERGY-ORIENTED DIFFUSION BRIDGE

We begin by framing image restoration through the lens of data geometry, positing that the task is fundamentally a transport problem between distinct data manifolds. Within this framework, the ideal restoration trajectory between a clean image $\mathbf{X}_0$ and its degraded version $\mathbf{Y}$ should be the geodesic $\gamma(t)$ on the underlying data manifold, representing the most efficient transformation. To regularize the diffusion trajectory following the bridge trajectory, we have formulated the following Proposition 4.1.

**Proposition 4.1** (Kinetic Energy Minimization of the Geodesic Expectation). *Let $X_t$ be a stochastic diffusion process defined over the controllable time horizon $t \in [0, T_0]$. If the process follows the manifold geodesic $\gamma(t)$, defined as the trajectory that minimizes the kinetic transport energy between the starting distribution $Y$ and the target distribution $X$, then the deterministic component (expectation) of the resulting trajectory, $\mu(t) = \mathbb{E}[\widetilde{\mathbf{X}}_t]$, must satisfy the following linear transport equation:*

$$\mathbb{E}[\widetilde{\mathbf{X}}_t] = \left[1 - \left(\frac{t}{T_0}\right)\right]\mathbf{X}_0 + \left(\frac{t}{T_0}\right)\mathbf{Y}. \tag{5}$$

*Proof.* Please refer to Appendix B.1. □

Based on that, we formulate a novel diffusion bridge defined for a controllable time horizon $t \in [0, T_0]$ with $T_0 \in (0, 1]$, which is formulated as:

$$\underbrace{\mathbf{X}_t = \alpha_t\widetilde{\mathbf{X}}_t + \beta_t\mathbf{X}_T,}_{\text{Diffusion Trajectory}} \quad \text{where} \quad \underbrace{\widetilde{\mathbf{X}}_t = \left[1 - \left(\frac{t}{T_0}\right)\right]\mathbf{X}_0 + \left(\frac{t}{T_0}\right)\mathbf{Y}.}_{\text{Inherent Data Geodesic Transition}} \tag{6}$$

However, $\alpha_t$ and $\beta_t$ are still under-defined. Given that we leverage the prior of the pretrained rectified flow neural network (Andreas et al., 2024) to boost the reconstruction performance, we further utilize this as a regularization to explore the optimal $\alpha_t$ and $\beta_t$ selection in Proposition 4.2.

**Proposition 4.2** (Minimal Adaptation Energy via Lipschitz Stability). *Let $\epsilon_\theta$ be a pretrained Rectified Flow network with Lipschitz constant $K$, originally trained on the trajectory $\mathbf{Z}_t = (1-t)\mathbf{Z}_0 + t\mathbf{Z}_1$. When fine-tuning this network on a new bridge process $\mathbf{X}_t = \alpha_t \tilde{\mathbf{X}}_t + \beta_t \mathbf{X}_T$, the Adaptation Energy, defined as the expected squared deviation in the network's output caused by trajectory mismatch, is minimized if and only if*

$$\alpha_t = 1 - t, \text{and} \quad \beta_t = t. \tag{7}$$

*Proof.* Please refer to Appendix B.2. □

These selections of $\alpha_t$ and $\beta_t$ ensure the process matches the linear signal-to-noise evolution inherent to Rectified Flow models. In the following selection, we introduce the consistency-based tuning to further boost the effectiveness and efficiency of the reconstruction algorithm.

## 4.2 CONSISTENCY E-BRIDGE SOLVER

Although the E-Bridge defined in Sec. 4.1 provides a theoretically efficient trajectory blueprint, it presents two subsequent challenges: first, how to traverse this trajectory without resorting to slow, iterative ODE solvers, and second, how to ensure this traversal remains faithful to the complex, underlying data manifold. To address these issues, we introduce a consistency E-Bridge solver, which is not an iterative sampler but a direct single-step mapping function. Its design is a synthesis of two core principles: a principled architecture derived from the analytical inversion of the E-Bridge trajectory equation and a powerful training objective based on continuous-time geodesic consistency.

**Solver Derivation via Manifold-Guided Trajectory Inversion.** A key theoretical insight of our work is that the endpoint $\mathbf{X}_0$ of the E-Bridge trajectory can be solved analytically. This requires reframing the problem from simulating a stochastic trajectory to solving a deterministic equation. The challenge lies in the fact that the noise term $\mathbf{X}_T$ is unknown. We resolve this by positing that for any point $\mathbf{X}_t$ on the trajectory, a pretrained flow-matching network $\epsilon_\theta(\cdot)$ can provide an instantaneous estimate of the vector pointing to the terminal noise:

$$\epsilon_\theta(\mathbf{X}_t) = \mathbf{X}_T - \mathbf{X}_0. \tag{8}$$

For any state $\mathbf{X}_t$ residing in the ambient space (potentially off the data manifold), the denoiser $\epsilon_\theta(\cdot)$ allows us to define an instantaneous projection onto the learned manifold of clean images. This projection, denoted as $\hat{\mathbf{X}}_0(\mathbf{X}_t)$, represents the denoiser's estimate of the clean image corresponding to $\mathbf{X}_t$. Consequently, the vector $\hat{\mathbf{X}}_0(\mathbf{X}_t) - \mathbf{X}_t$ forms a correction field, providing the precise direction needed to steer the state $\mathbf{X}_t$ back towards the manifold. It is this ability to provide a geodesic transition that makes $\epsilon_\theta(\cdot)$ the ideal tool for our trajectory inversion.

By substituting the relation from Eq. (8) into our E-Bridge trajectory definition in Eq. (6), we leverage this geometric correction to transform the trajectory into a single implicit algebraic equation where $\mathbf{X}_0$ is the sole unknown:

$$\mathbf{X}_t = (1-t)\left[\left(1 - \left(\frac{t}{T_0}\right)\right)\mathbf{X}_0 + \left(\frac{t}{T_0}\right)\mathbf{Y}\right] + t(\epsilon_\theta(\mathbf{X}_t) + \mathbf{X}_0). \tag{9}$$

The linearity of this equation with respect to $\mathbf{X}_0$ permits a closed-form solution via algebraic inversion. This yields the principled architecture of our E-Bridge solver, $\mathcal{F}_\theta(\cdot)$:

$$\hat{\mathbf{X}}_0 := \mathcal{F}_\theta(\mathbf{X}_t, \mathbf{Y}, t) = \frac{\mathbf{X}_t - A(t)\mathbf{Y} - B(t)\epsilon_\theta(\mathbf{X}_t)}{C(t)}, \tag{10}$$

where the coefficients $A(t) = (1-t)\left(\frac{t}{T_0}\right)$, $B(t) = t$, $C(t) = 1 - (1-t)\left(\frac{t}{T_0}\right)$ are derived directly from the algebraic manipulation of Eq. (9). This formulation reveals the true power of our E-Bridge model: it is not merely an interpolator, but a direct solver for the endpoint of a geometrically motivated trajectory.

**Training as Energy-oriented Geodesic Consistency Enforcement.** The training objective is endowed with a clear geometric meaning. We are not just forcing a network to be self-consistent; we are enforcing energy-oriented geodesic consistency. This principle asserts that for any point on a given geodesic trajectory, a perfect solver must map it to the same, invariant endpoint $\mathbf{X}_0$.

---

**Algorithm 1:** Training Process of Energy-oriented Diffusion Bridge

---

**Input** : Diffusion model parameter $\boldsymbol{\theta}$, dataset distribution $P(\mathbf{X}_0, \mathbf{Y})$; Hyperparameters: trajectory length $T_0$.

**Output :** Updated parameter $\boldsymbol{\theta}$.

1 Initialize parameters $\theta$ with the pretrained diffusion model.

2 **while** *not converged* **do**

    // Sample data pairs, timestep, and noise.

3    $\{\mathbf{X}_0, \mathbf{Y}\} \sim P(\mathbf{X}_0, \mathbf{Y})$, $t \sim \mathcal{U}(0, T_0]$, $\mathbf{X}_T \sim \mathcal{N}(0, \mathbf{I})$;

    // Initialize state interpolation by Eq. (6)

4    $\widetilde{\mathbf{X}}_t \leftarrow \left[1 - \left(\frac{t}{T_0}\right)\right] \mathbf{X}_0 + \left(\frac{t}{T_0}\right) \mathbf{Y}$;

5    $\mathbf{X}_t \leftarrow (1-t)\widetilde{\mathbf{X}}_t + t\mathbf{X}_T$;

    // Estimate clean image through Eq. (10)

6    $\widehat{\mathbf{X}}_0 \leftarrow \mathcal{F}_\theta(\mathbf{X}_t, \mathbf{Y}, t)$;

    // Model update based on consistency optimization in Eq. (11)

7    $\boldsymbol{\theta} \leftarrow \boldsymbol{\theta} - \nabla_{\boldsymbol{\theta}}\langle\widehat{\mathbf{X}}_0, \frac{d\mathcal{F}_{\theta-}(\mathbf{X}_t, \mathbf{Y}, t)}{dt}\rangle$;

**Return:** $\boldsymbol{\theta}$.

---

In the continuous-time domain, this is equivalent to requiring the solver's output to have a total derivative of zero along the path. We enforce this directly by adopting the continuous-time consistency objective (Song et al., 2023; Lu & Song, 2025).

Instead of a discrete distance metric, our training objective is defined by its gradient with respect to the model parameters $\theta$. Following the formulation of continuous-time consistency models, this gradient is given by:

$$\nabla_\theta \mathcal{L}_{\text{E-Bridge}}(\theta) = \mathbb{E}_{\mathbf{X}_t, \mathbf{Y}, t}\left[\nabla_\theta\left(\mathcal{F}_\theta(\mathbf{X}_t, \mathbf{Y}, t)^\top \cdot \text{sg}\left(\frac{d\mathcal{F}_{\theta-}(\mathbf{X}_t, \mathbf{Y}, t)}{dt}\right)\right)\right], \qquad (11)$$

where $\frac{dF_{\theta-}}{dt}$ is the total derivative of the target solver's output along the geodesic trajectory, and $\text{sg}(\cdot)$ is the stop-gradient operator.

This objective enforces geodesic consistency in the continuous domain. The term $\text{sg}(\frac{d\mathcal{F}_{\theta-}}{dt})$ acts as an inconsistency tangent vector; it measures the instantaneous "drift" of the solver's output at time $t$. The optimization process minimizes the inner product between the online solver's output $\mathcal{F}_\theta$ and this inconsistency vector. The only stable equilibrium for this process across the entire trajectory is for the inconsistency vector itself to become zero. This elegantly forces the solver to satisfy the condition $\frac{d\mathcal{F}_\theta}{dt} = \mathbf{0}$, ensuring its output remains constant and thus adheres to the principle of geodesic consistency.

By optimizing this continuous objective, we train the underlying network $\epsilon_\theta$ to accurately capture the local geometry of the data manifold. This, in turn, allows the solver $\mathcal{F}_\theta$ to function as an effective and efficient geodesic solver. This training enables both high-fidelity single-step restoration and the possibility of iterative refinement along the learned optimal path. The training process is illustrated in Algorithm 1.

### 4.3 TRAJECTORY ADAPTIVITY OF E-BRIDGE FOR IMAGE RESTORATION

A significant, yet often unaddressed, limitation of many diffusion-based restoration methods is their reliance on a one-size-fits-all restoration path. Whether for denoising, super-resolution, or other tasks, these models typically employ a fixed diffusion trajectory (e.g., a reverse process from $t = 1$ to $t = 0$). This approach ignores the heterogeneous nature of image degradations: the information gap between a noisy image and its clean version is vastly different from the gap between a low-quality image and its high-quality target. Our E-Bridge framework provides a principled mechanism for addressing this through the dynamic trajectory length parameter, $T_0$, which is more than just a trajectory shortener; it is a powerful task-adaptive control knob that adapts the restoration process to the specific nature of the input degradation. The choice of $T_0$ directly determines the composition of the restoration's starting point, $\mathbf{X}_{T_0} = (1 - T_0)\mathbf{Y} + T_0\mathbf{X}_T$. This allows us to precisely balance

Table 1: Quantitative comparisons of different diffusion-based methods across multiple image restoration tasks. The best results are highlighted in **bold**. † represents that we utilized the official public code and recommended settings to train the baseline on the same training dataset as ours.

| Task | Method | NFE↓ | PSNR↑ | LPIPS↓ | FID↓ | NIQE↓ | MUSIQ↑ |
|---|---|---|---|---|---|---|---|
| Super-resolution | DiffBIR (Lin et al., 2024) | 50 | 22.601 | 0.404 | 77.767 | 4.333 | 64.767 |
| | AutoDIR (Jiang et al., 2024) | 25 | 23.953 | 0.419 | 64.995 | 5.599 | 48.103 |
| | UniDB† (Zhu et al., 2025a) | 100 | 20.778 | 0.477 | 87.135 | 3.882 | 62.004 |
| | IRBridge† (Wang et al., 2025a) | 100 | 23.615 | 0.415 | 59.539 | 5.215 | 65.535 |
| | MaRS† (Li et al., 2025) | 20 | 23.325 | 0.525 | 104.665 | 6.328 | 53.162 |
| | MaRS† (Li et al., 2025) | 10 | 23.786 | 0.539 | 101.978 | 7.251 | 49.489 |
| | UniDB++† (Pan et al., 2025) | 10 | 23.383 | 0.469 | 79.383 | 5.709 | 57.237 |
| | UniDB++† (Pan et al., 2025) | 5 | **24.393** | 0.494 | 70.664 | 5.904 | 63.087 |
| | E-Bridge (Ours) | 10 | 21.282 | **0.346** | 57.837 | **3.839** | **70.868** |
| | E-Bridge (Ours) | 5 | 22.477 | 0.356 | **55.236** | 4.373 | 65.342 |
| | UniDB++† (Pan et al., 2025) | 1 | 23.983 | 0.658 | 83.718 | 7.596 | 33.222 |
| | E-Bridge (Ours) | 1 | 24.094 | 0.452 | 72.001 | 6.251 | 44.605 |
| Denoising | DA-CLIP (Luo et al., 2024) | 100 | 27.027 | 0.301 | 39.526 | 5.091 | 58.098 |
| | AutoDIR (Jiang et al., 2024) | 25 | 28.063 | 0.282 | 28.452 | 5.095 | 58.399 |
| | UniDB† (Zhu et al., 2025a) | 100 | 27.956 | 0.212 | 28.256 | **3.019** | 63.782 |
| | IRBridge† (Wang et al., 2025a) | 100 | 25.696 | 0.273 | 41.936 | 4.376 | 69.053 |
| | MaRS† (Li et al., 2025) | 20 | 28.465 | 0.220 | 29.338 | 3.993 | 63.004 |
| | MaRS† (Li et al., 2025) | 10 | 28.252 | 0.293 | 33.204 | 5.185 | 57.677 |
| | UniDB++† (Pan et al., 2025) | 10 | 28.515 | 0.224 | 33.310 | 4.144 | 68.863 |
| | UniDB++† (Pan et al., 2025) | 5 | **29.308** | 0.214 | 34.872 | 4.394 | 69.048 |
| | E-Bridge (Ours) | 10 | 26.069 | 0.184 | 26.611 | 3.514 | **69.736** |
| | E-Bridge (Ours) | 5 | 26.649 | **0.182** | **26.286** | 4.492 | 67.507 |
| | UniDB++† (Pan et al., 2025) | 1 | 21.844 | 0.613 | 92.362 | 8.197 | 45.150 |
| | E-Bridge (Ours) | 1 | 25.241 | 0.258 | 36.967 | 4.535 | 58.954 |
| Raindrop Removal | DA-CLIP (Luo et al., 2024) | 100 | 31.241 | **0.061** | **24.066** | 4.817 | 67.593 |
| | AutoDIR (Jiang et al., 2024) | 50 | **32.332** | 0.092 | 26.789 | 3.365 | 68.723 |
| | UniDB† (Zhu et al., 2025a) | 100 | 30.344 | 0.071 | 25.570 | 4.736 | 67.599 |
| | IRBridge† (Wang et al., 2025a) | 100 | 28.771 | 0.125 | 54.814 | 3.959 | 65.012 |
| | MaRS† (Li et al., 2025) | 20 | 31.664 | 0.070 | 24.554 | 5.833 | 66.249 |
| | MaRS† (Li et al., 2025) | 10 | 30.359 | 0.167 | 41.186 | 7.749 | 59.468 |
| | UniDB++† (Pan et al., 2025) | 10 | 32.151 | 0.068 | 26.184 | 3.577 | 69.761 |
| | UniDB++† (Pan et al., 2025) | 5 | 32.327 | 0.072 | 28.866 | 3.515 | **70.498** |
| | E-Bridge (Ours) | 10 | 29.222 | 0.079 | 30.551 | **3.221** | 70.006 |
| | E-Bridge (Ours) | 5 | 29.611 | 0.080 | 29.726 | 3.449 | 67.379 |
| | UniDB++† (Pan et al., 2025) | 1 | 25.420 | 0.174 | 61.693 | 3.683 | 64.523 |
| | E-Bridge (Ours) | 1 | 29.220 | 0.085 | 33.408 | 3.322 | 67.301 |
| Low-light | DA-CLIP (Luo et al., 2024) | 100 | 23.091 | 0.117 | 39.428 | 4.359 | 71.245 |
| | AutoDIR (Jiang et al., 2024) | 50 | **26.961** | 0.098 | 38.118 | 5.208 | 70.500 |
| | IRBridge† (Wang et al., 2025a) | 100 | 23.913 | 0.116 | 41.007 | 4.664 | 64.926 |
| | DiffUIR (Zheng et al., 2024) | 3 | 25.160 | 0.160 | 71.703 | 4.904 | 71.390 |
| | E-Bridge (Ours) | 10 | 25.538 | **0.097** | 41.197 | **4.198** | **71.684** |
| | E-Bridge (Ours) | 5 | 25.847 | 0.107 | 38.996 | 4.211 | 70.868 |
| | AdaIR (Cui et al., 2025) | 1 | 22.634 | 0.168 | 58.106 | 4.713 | 70.901 |
| | E-Bridge (Ours) | 1 | 24.656 | 0.133 | 54.787 | 4.331 | 66.805 |
| Demoiréing | UniDB† (Zhu et al., 2025a) | 100 | 19.719 | 0.283 | 61.063 | 5.768 | 61.031 |
| | IRBridge† (Wang et al., 2025a) | 100 | **21.441** | 0.277 | 36.367 | 6.158 | 61.199 |
| | MaRS† (Li et al., 2025) | 20 | 19.965 | 0.239 | 40.067 | 5.697 | 60.894 |
| | MaRS† (Li et al., 2025) | 10 | 20.118 | 0.346 | 41.186 | 7.679 | 55.420 |
| | UniDB++† (Pan et al., 2025) | 10 | 20.109 | 0.191 | 37.558 | 5.416 | 61.466 |
| | UniDB++† (Pan et al., 2025) | 5 | 20.527 | **0.183** | 37.335 | 5.452 | 62.054 |
| | E-Bridge (Ours) | 10 | 20.263 | 0.313 | 42.973 | **5.223** | **63.932** |
| | E-Bridge (Ours) | 5 | 20.849 | 0.230 | 36.194 | 5.437 | 62.155 |
| | UniDB++† (Pan et al., 2025) | 1 | 20.181 | 0.266 | 49.057 | 5.603 | 55.618 |
| | E-Bridge (Ours) | 1 | 20.715 | 0.254 | **34.414** | 5.342 | 56.788 |

the trade-off between preserving information from the degraded input and leveraging the generative power of the diffusion model. We call this the information-entropy trade-off.

For mild degradations like denoising, the degraded image $\mathbf{Y}$ contains a substantial amount of reliable high-frequency structure and is already a strong prior for $\mathbf{X}_0$. Here, we desire high *information preservation*. By choosing a small $T_0$, the starting point $\mathbf{X}_{T_0}$ remains very close to $\mathbf{Y}$. The restoration process becomes a short, targeted refinement path, correcting deviations without unnecessarily destroying the trustworthy information already present.

For severe degradations like super-resolution, $\mathbf{Y}$ is a poor prior for $\mathbf{X}_0$. It lacks fundamental high-frequency details, and its structure may contain misleading artifacts. Relying heavily on it would

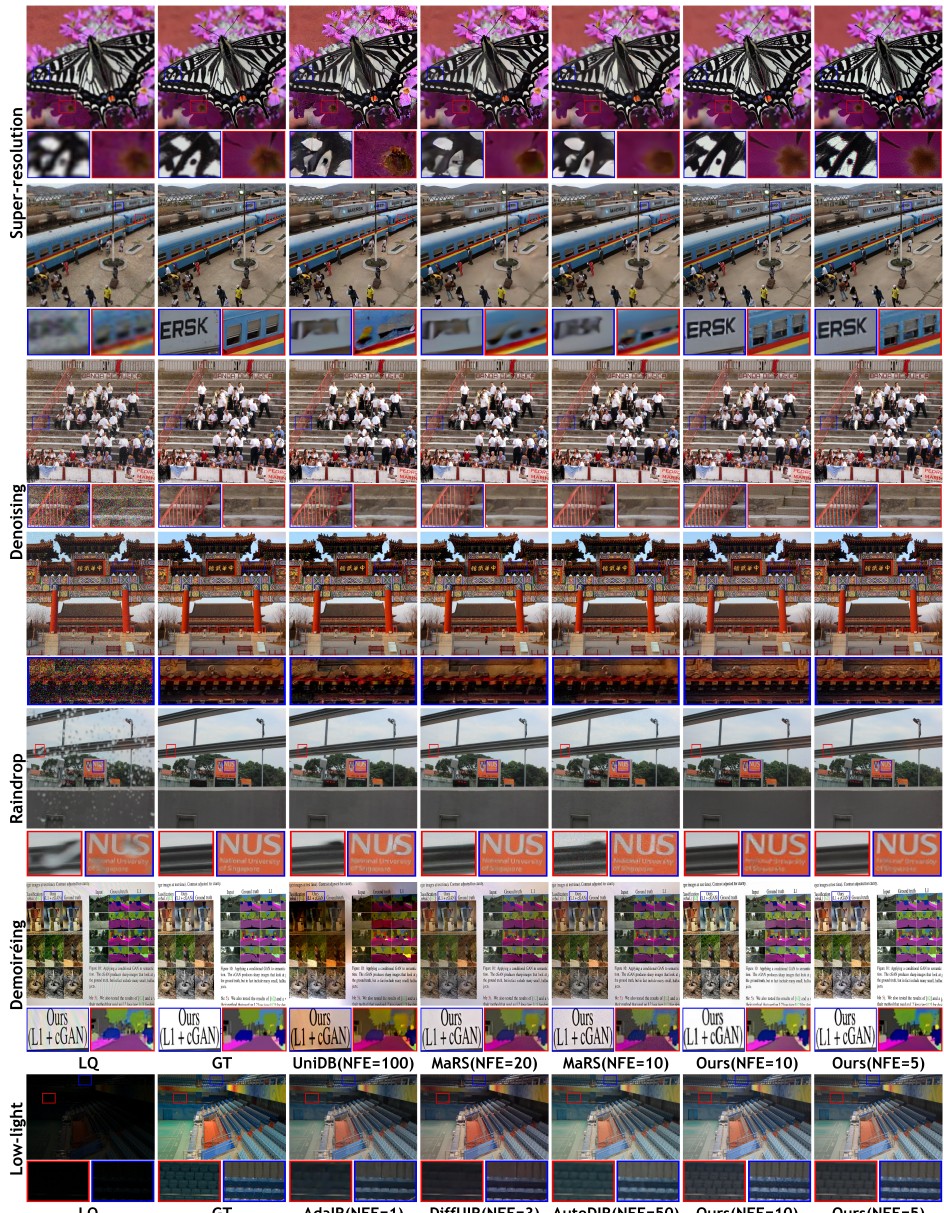

Figure 2: Visual comparison of different methods across various tasks.

cap the restoration quality. Here, we need high *generative power*. By choosing a large $T_0$ (e.g., $T_0 \to 1$), the starting point $\mathbf{X}_{T_0}$ becomes dominated by Gaussian noise, effectively erasing the unreliable details of $\mathbf{Y}$ while retaining it as a faint structural guide. This provides the model with a longer, higher-entropy path, giving it the necessary room to hallucinate complex, realistic details from scratch, conditioned on the low-frequency cues from $\mathbf{Y}$.

# 5 EXPERIMENTS

## 5.1 EXPERIMENT SETTINGS

**Datasets.** We conducted a comprehensive evaluation of our proposed method across five challenging image restoration tasks, including super-resolution, denoising, raindrop removal, low-light image enhancement, and image demoiréing. Specifically, for super-resolution, we trained on 3,450 images from DIV2K (Agustsson & Timofte, 2017) and Flickr2K (Lim et al., 2017), with degrada-

Table 2: Ablation study on the impact of the proposed geodesic trajectory. Specifically, we replaced the geodesic trajectory with two other types of trajectories, i.e., **a**) the standard diffusion trajectory, starting from pure Gaussian noise to the clean image manifold, and **b**) the conventional bridge trajectory, which constructs a trajectory from the degraded to the clean image, often follows a "re-noising" phase before denoising. Best results in each row are in **bold**.

| Task | NFE | (a) | | | | (b) | | | | Ours | | | |
|---|---|---|---|---|---|---|---|---|---|---|---|---|---|
| | | PSNR↑ | LPIPS↓ | MUSIQ↑ | NIQE↓ | PSNR↑ | LPIPS↓ | MUSIQ↑ | NIQE↓ | PSNR↑ | LPIPS↓ | MUSIQ↑ | NIQE↓ |
| Super-resolution | 10 | 20.527 | 0.464 | 63.476 | 3.973 | 19.945 | 0.446 | 69.157 | 4.958 | **21.282** | **0.346** | **70.868** | **3.839** |
| | 5 | 21.016 | 0.495 | 56.709 | 5.657 | 21.443 | 0.502 | 63.068 | **3.902** | **22.477** | **0.356** | **65.342** | 4.373 |
| | 1 | 19.667 | 0.486 | 31.919 | 7.166 | 22.039 | 0.481 | 35.552 | **4.267** | **24.094** | **0.452** | **44.605** | 6.251 |
| Denoising | 10 | 24.241 | 0.288 | 64.724 | 4.952 | 24.391 | 0.261 | **69.976** | 5.575 | **26.069** | **0.184** | 69.736 | **3.514** |
| | 5 | 23.701 | 0.316 | 60.090 | 5.235 | 25.279 | 0.235 | 65.870 | **3.807** | **26.649** | **0.182** | **67.507** | 4.492 |
| | 1 | 21.693 | 0.460 | 45.747 | 5.889 | 24.516 | 0.371 | 51.381 | 4.552 | **25.241** | **0.258** | **58.954** | **4.535** |
| Raindrop | 10 | 24.919 | 0.136 | 68.559 | 3.229 | 26.346 | 0.148 | 67.546 | 3.428 | **29.222** | **0.079** | **70.006** | **3.221** |
| | 5 | 24.702 | 0.155 | 66.448 | 4.023 | 27.471 | 0.118 | 64.260 | 3.462 | **29.611** | **0.080** | **67.379** | **3.449** |
| | 1 | 23.990 | 0.319 | 52.946 | 6.189 | 26.202 | 0.252 | 55.173 | 3.528 | **29.220** | **0.085** | **67.301** | **3.322** |
| Low-light | 10 | 23.375 | 0.144 | 68.498 | **3.881** | 22.176 | 0.181 | 70.109 | 4.234 | **25.538** | **0.097** | **71.684** | 4.198 |
| | 5 | 22.706 | 0.145 | 66.551 | **3.746** | 22.316 | 0.157 | 69.690 | 4.300 | **25.847** | **0.107** | **70.868** | 4.211 |
| | 1 | 21.634 | 0.333 | 48.405 | 4.365 | 23.661 | 0.358 | 52.948 | 4.399 | **24.656** | **0.133** | **66.805** | **4.331** |
| Demoiréing | 10 | 19.260 | 0.344 | 60.277 | 5.628 | 19.956 | **0.252** | 60.216 | 5.697 | **20.263** | 0.313 | **63.932** | **5.223** |
| | 5 | 19.601 | 0.429 | 56.406 | 6.342 | 20.076 | 0.240 | 59.877 | **5.161** | **20.849** | **0.230** | **62.155** | 5.437 |
| | 1 | 19.301 | 0.469 | 44.049 | 6.840 | 19.468 | 0.372 | 51.601 | 5.776 | **20.715** | **0.254** | **56.788** | **5.342** |

tions generated by Real-ESRGAN (Wang et al., 2021). We tested on 100 images from the DIV2K validation set. For the denoising task, we utilized the same high-quality images from the SR task, synthetically corrupted with Gaussian noise ($\sigma = 50$). For low-light enhancement, we employed the LOLv1 (Wei et al., 2018). We used the UHDM dataset (Yu et al., 2022) for image demoiréing, with a split of 4,500 training and 500 testing images.

**Implementation details.** We constructed our E-Bridge framework based on the large-scale FLUX-dev model (Andreas et al., 2024). The model was then fine-tuned for 10,000 iterations using the Adam optimizer with a learning rate of $2 \times 10^{-5}$. We used a training patch size of $1024 \times 1024$ and an effective batch size of 128, achieved via gradient accumulation. The trajectory horizon $T_0$ was sampled from the interval $[0.2, 0.95]$ during training, which endows the model with strong adaptability. Consequently, at inference time, $T_0$ becomes a tunable knob that can be set on demand to precisely control the restoration process. This allows us to select a smaller $T_0$ for tasks requiring high fidelity and a larger $T_0$ for tasks demanding stronger generative power. All experiments were conducted on NVIDIA A800 GPUs.

**Evaluation metrics.** For quantitative evaluation, we employ a comprehensive suite of metrics: full-reference scores such as Peak Signal-to-Noise Ratio (PSNR) within the YCbCr color space and LPIPS; no-reference perceptual scores including NIQE (Mittal et al., 2012), and MUSIQ (Ke et al., 2021); and the distributional metric FID. We also report the Number of Function Evaluations (NFE) to measure computational efficiency. To leverage the full generative capacity of the pre-trained FLUX model and account for the ill-posed nature of image restoration, we avoid overly constraining the output to the ground-truth reference. Consequently, our evaluation relies heavily on non-reference and perceptual scores such as NIQE and MUSIQ.

## 5.2 COMPARISON WITH STATE-OF-THE-ART METHODS

As presented in Table 1, our proposed E-Bridge consistently achieves state-of-the-art or highly competitive performance on a comprehensive suite of perceptual and distributional metrics, including LPIPS, FID, NIQE, and MUSIQ. Crucially, it accomplishes this with a remarkable reduction in computational cost, requiring only 5 to 10 NFE. This highlights the effectiveness and efficiency of our geodesic trajectory and consistency-based solver. Although not always leading in PSNR, our approach consciously prioritizes perceptual realism over pixel-wise accuracy. Methods that narrowly optimize for PSNR are known to produce overly smoothed results that lack convincing high-frequency details. Our framework, by contrast, is capable of generating perceptually rich images. As visually substantiated in Figure 2, our method can even produce results with sharper details and more pleasing textures than the original GT, which may contain slight softness or compression artifacts. This underscores E-Bridge's ability to generate an idealized, visually plausible outcome rather than simply mimicking a potentially imperfect ground-truth.

Table 3: Comprehensive efficiency comparison between our consistency-based solver (E-Bridge) and a traditional iterative ODE solver (DDIM). (P: PSNR↑, L: LPIPS↓, M: MUSIQ↑, N: NIQE↓). Best results in each column are in **bold**.

| Method | NFE | Denoising P↑ | L↓ | M↑ | N↓ | Super-res. P↑ | L↓ | M↑ | N↓ | Raindrop rem. P↑ | L↓ | M↑ | N↓ | Low-light Enh. P↑ | L↓ | M↑ | N↓ | Demoireing P↑ | L↓ | M↑ | N↓ |
|---|---|---|---|---|---|---|---|---|---|---|---|---|---|---|---|---|---|---|---|---|---|
| DDIM | 50 | 26.31 | **0.176** | 69.01 | 3.55 | 21.73 | **0.330** | 69.37 | **3.26** | 29.20 | 0.090 | 68.81 | **3.14** | 25.37 | 0.112 | 71.48 | **4.08** | 20.15 | **0.221** | 62.80 | **5.15** |
|  | 20 | 26.53 | 0.185 | 68.79 | 3.94 | 21.99 | 0.353 | 68.85 | 3.86 | 29.51 | 0.086 | 67.10 | 3.27 | 25.45 | 0.112 | 70.83 | 4.19 | 20.31 | 0.303 | 62.35 | 5.31 |
|  | 10 | **26.71** | 0.195 | 67.44 | 4.58 | 22.27 | 0.362 | 67.17 | 5.02 | 29.72 | 0.086 | 66.46 | 3.57 | 25.57 | 0.111 | 70.61 | 4.32 | 20.43 | 0.327 | 62.02 | 5.50 |
|  | 1 | 24.56 | 0.270 | 57.07 | 4.30 | 17.92 | 0.591 | 46.07 | 8.97 | 27.48 | 0.135 | 62.42 | 4.39 | 20.96 | 0.177 | 63.78 | 4.27 | 17.44 | 0.407 | 44.21 | 5.33 |
| Ours | 10 | 26.06 | 0.184 | **69.74** | **3.51** | 21.28 | 0.346 | **70.87** | 3.84 | 29.22 | **0.079** | **70.01** | 3.22 | 25.54 | **0.097** | 71.68 | 4.20 | 20.26 | 0.313 | **63.93** | 5.22 |
|  | 5 | 26.65 | 0.182 | 67.51 | 4.49 | **22.48** | 0.356 | 65.34 | 4.37 | **29.61** | 0.080 | 67.38 | 3.45 | **25.85** | 0.107 | 70.87 | 4.21 | **20.85** | 0.230 | 62.16 | 5.44 |
|  | 1 | 25.24 | 0.258 | 58.95 | 4.54 | 24.09 | 0.452 | 44.61 | 6.25 | 29.22 | 0.085 | 67.30 | 3.32 | 24.66 | 0.133 | 64.81 | 4.33 | 20.72 | 0.254 | 56.79 | 5.34 |

Table 4: Analysis of the trajectory horizon parameter $T_0$. (P: PSNR↑, L: LPIPS↓, M: MUSIQ↑, N: NIQE↓). Best results for each task are in **bold**.

| Task | $T_0 = 0.3$ P↑ | L↓ | M↑ | N↓ | $T_0 = 0.5$ P↑ | L↓ | M↑ | N↓ | $T_0 = 0.7$ P↑ | L↓ | M↑ | N↓ | $T_0 = 0.9$ P↑ | L↓ | M↑ | N↓ |
|---|---|---|---|---|---|---|---|---|---|---|---|---|---|---|---|---|
| Super-resolution | 21.12 | 0.457 | 62.16 | 4.88 | 21.49 | 0.397 | 67.00 | 4.36 | **21.82** | 0.359 | 68.77 | 3.84 | 21.28 | **0.346** | **70.87** | **3.84** |
| Denoising | 24.43 | 0.276 | 66.13 | 4.48 | 25.35 | 0.206 | 69.24 | 3.74 | 26.06 | **0.184** | **69.73** | **3.51** | **26.53** | 0.185 | 69.13 | 4.04 |
| Low-light | 24.89 | 0.129 | 68.55 | **3.91** | 25.12 | 0.118 | 70.93 | 4.07 | 25.35 | 0.107 | 71.39 | 4.15 | **25.54** | **0.097** | **71.68** | 4.20 |
| Raindrop removal | 29.61 | **0.080** | 67.38 | 3.45 | 29.74 | 0.081 | 66.73 | 3.57 | 29.70 | 0.083 | 65.23 | 3.821 | 29.21 | 0.092 | 63.98 | 3.97 |
| Demoiréing | 20.85 | 0.234 | 61.67 | **5.34** | 20.85 | **0.230** | **62.16** | 5.44 | 19.76 | 0.249 | 61.32 | 5.60 | 16.47 | 0.328 | 59.04 | 5.68 |

## 5.3 ABLATION STUDY

**Effectiveness of the E-Bridge.** We compare our geodesic trajectory with two baselines from Fig. 1. It can be observed that our method consistently outperforms both alternatives in perceptual metrics (LPIPS, MUSIQ, NIQE) and fidelity (PSNR) across all five image restoration tasks. These results confirm that the proposed low-energy geodesic path significantly enhances restoration quality compared to sub-optimal traditional trajectories.

**Effectiveness of the Consistency Solver.** As shown in Table 3, our consistency-based solver demonstrates overwhelming efficiency compared to the traditional iterative solver. For denoising, our E-Bridge achieves the best perceptual quality (MUSIQ/NIQE) with only 10 NFE, surpassing the iterative solver, which requires 50 NFE. For super-resolution, the advantage is even more pronounced: our E-Bridge not only secures the highest generative quality (MUSIQ) at 10 NFE but also achieves the best fidelity (PSNR) with just 5 NFE. Overall, our E-Bridge delivers superior or competitive performance at a fraction of the computational cost (5-10 NFE vs. 50 NFE), proving its fundamental efficiency across different tasks.

**Analysis of the Trajectory Horizon $T_0$.** We investigate the role of the trajectory horizon $T_0$ as a task-adaptive control knob, with results presented in Table 4. This analysis validates our hypothesis that $T_0$ effectively balances the trade-off between information preservation and generative power, as outlined in Sec. 4.3. For the denoising task, a moderate $T_0 = 0.7$ provides the best perceptual quality. Conversely, for the super-resolution task, a larger $T_0 = 0.9$ is required to unlock the model's generative potential, yielding the best perceptual scores (LPIPS and MUSIQ). For tasks such as raindrop removal and demoiréing, where the degraded images retain significant useful information, smaller values of $T_0$ are preferable. This study confirms that $T_0$ is a critical control parameter that allows the restoration process to be precisely adapted to the demands of different degradations, balancing information preservation against generative power.

## 6 CONCLUSION

We have presented the Energy-oriented diffusion Bridge (E-Bridge), a new framework that sets a benchmark for efficient, high-fidelity image restoration. By constructing a low-energy geodesic trajectory and deriving a single-step consistency solver, our method overcomes the core limitations of previous diffusion bridge models. The proposed E-Bridge framework avoids redundant re-noising phases and circumvents the need for slow iterative sampling, enabling high-quality restoration with fewer sampling steps. Extensive experiments demonstrate that our E-Bridge achieves state-of-the-art perceptual quality across various tasks.

ACKNOWLEDGMENTS

This work was supported in part by the National Natural Science Foundation of China under Grant 62422118, and in part by the Hong Kong Research Grants Council under Grant 11218121 and Grant N_CityU1114/25.

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

CONTENTS

## A APPENDIX OVERVIEW

This appendix is organized as follows. In Sec. B, we present rigorous proofs and analyses of the proposed method. Specifically, Secs. B.1 and B.2 demonstrate the parameterization process of the proposed SDEs with the energy-based criteria, which establishes both transportation and control energy superiority of the proposed method. Then, we also validate that the designed scheme is geodesic and energy-optimal in Secs. B.3 and B.4. Building on these energy-based criteria, we then validate the advantages of the proposed energy-oriented geodesic bridge over the Ornstein–Uhlenbeck Process in Sec. C. Because our method has the same trajectory expectation as the Ornstein–Uhlenbeck Process, the control energy is the same. We only analyze the transportation energy. Finally, Sec. D provides further analysis of the Schrödinger diffusion bridge.

## B ENERGY-ORIENTED DIFFUSION BRIDGE

### B.1 PROOF OF ENERGY-DERIVED GEOMETRIC LINEARITY

Our proof formally derives the linear interpolation formula $\mathbb{E}[\mathbf{X}_t] = (1 - t/T_0)\mathbf{Y} + (t/T_0)\mathbf{X}$ by demonstrating that it is the unique solution to the geodesic energy minimization problem.

The proof proceeds by establishing that the geodesic $\gamma(t)$ on the data manifold corresponds to the path of constant velocity, and subsequently solving for the specific trajectory given the boundary conditions $\mathbf{Y}$ and $\mathbf{X}$.

**Variational Definition of the Geodesic**. Let $\mathcal{M}$ be the data manifold. The geodesic $\gamma(t)$ connecting the starting distribution $\mathbf{Y}$ (at $t = 0$) and the target distribution $\mathbf{X}$ (at $t = T_0$) is defined as the curve that minimizes the kinetic energy functional:

$$E[\gamma] = \frac{1}{2} \int_0^{T_0} ||\dot{\gamma}(t)||^2 dt \tag{12}$$

**Energy Minimization via Jensen's Inequality**. To find the optimal trajectory $\mu(t) = \mathbb{E}[\mathbf{X}_t]$ that coincides with this geodesic, we analyze the energy required to traverse the displacement. By applying Jensen's Inequality to the convex function $f(x) = \|x\|^2$ over the interval $[0, T_0]$:

$$\frac{1}{T_0} \int_0^{T_0} \|\dot{\mu}(t)\|^2 dt \geq \left\| \frac{1}{T_0} \int_0^{T_0} \dot{\mu}(t) dt \right\|^2. \tag{13}$$

The term inside the norm on the RHS is the average velocity, which is determined solely by the endpoints:

$$\frac{1}{T_0} \int_0^{T_0} \dot{\mu}(t) dt = \frac{\mu(T_0) - \mu(0)}{T_0} = \frac{\mathbf{Y} - \mathbf{X}_0}{T_0}. \tag{14}$$

Thus, the lower bound for the energy is:

$$E[\mu] \geq \frac{T_0}{2} \left\| \frac{\mathbf{Y} - \mathbf{X}_0}{T_0} \right\|^2 = \frac{\|\mathbf{Y} - \mathbf{X}_0\|^2}{2T_0}. \tag{15}$$

Equality holds if and only if the velocity $\dot{\mu}(t)$ is constant throughout $t \in [0, T_0]$.

**Derivation of the Deterministic Trajectory**. Let this constant velocity be $v^*$. Since the velocity is constant, we have $v^* = \frac{\mathbf{Y} - \mathbf{X}_0}{T_0}$. The trajectory is obtained by integrating the velocity from the starting point:

$$\mu(t) = \mu(0) + \int_0^t v^* d\tau = \mathbf{X}_0 + t \cdot \left( \frac{\mathbf{Y} - \mathbf{X}_0}{T_0} \right). \tag{16}$$

Rearranging the terms yields the linear interpolation formula stated in the proposition:

$$\mu(t) = \left( 1 - \frac{t}{T_0} \right) \mathbf{X}_0 + \left( \frac{t}{T_0} \right) \mathbf{Y}. \tag{17}$$

**Regularization via Finite Horizon** $T_0$. As discussed in Appendix E, the introduction of the explicit horizon $T_0$ is critical for ensuring the stability of the transport. In standard Stochastic Interpolant frameworks (e.g., Brownian Bridge) defined over $t \in [0, 1]$, the velocity field typically scales as $v_t(x) \propto \frac{1}{1-t}$. As $t \to 1$, this leads to a singularity where $\|v_t\| \to \infty$, resulting in an unbounded Lipschitz constant and unstable training.

By defining the geodesic transport over a controllable horizon $[0, T_0]$, we ensure that the velocity $v^* = \frac{\mathbf{Y} - \mathbf{X}_0}{T_0}$ remains strictly finite and bounded (provided $T_0 > 0$). Consequently, the kinetic energy of the trajectory is finite:

$$E_{min} = \frac{1}{2} \int_0^{T_0} \left\| \frac{\mathbf{Y} - \mathbf{X}_0}{T_0} \right\|^2 dt = \frac{1}{2} \frac{\|\mathbf{Y} - \mathbf{X}_0\|^2}{T_0} < \infty. \tag{18}$$

Thus, the linear interpolation derived above represents the unique, energetically optimal, and numerically stable geodesic connecting $\mathbf{X}_0$ and $\mathbf{Y}$.

## B.2 PROOF OF MINIMAL ADAPTATION ENERGY VIA LIPSCHITZ STABILITY

**Definition of the Adaptation Energy Functional**. We define the Adaptation Energy $\mathcal{E}$ as the expected $L^2$ cost incurred by the displacement between the target distribution of the pretrained model ($\mathbf{Z}_t$) and the input distribution of the new task ($\mathbf{X}_t$). This quantifies the "spectral gap" the fine-tuning process must bridge:

$$\mathcal{E}(\alpha, \beta) := \int_0^1 \mathbb{E} \left[ \|\epsilon_\theta(\mathbf{X}_t, t) - \epsilon_\theta(\mathbf{Z}_t, t)\|^2 \right] dt \tag{19}$$

where: $\mathbf{Z}_t = (1-t)\mathbf{Z}_0 + t\mathbf{Z}_1$ is the canonical Rectified Flow trajectory. $\mathbf{X}_t = \alpha_t \tilde{\mathbf{X}}_t + \beta_t \mathbf{X}_T$ is the proposed bridge trajectory. We assume asymptotic alignment of domains: $\tilde{\mathbf{X}}_t \sim \mathbf{Z}_0$ (clean/structure) and $\mathbf{X}_T \sim \mathbf{Z}_1$ (noise) in distribution.

By the **Lipschitz continuity** of the neural network $\epsilon_\theta$ with respect to its input state $\mathbf{X}$, we have the inequality:

$$\|\epsilon_\theta(x) - \epsilon_\theta(z)\|_2 \le K\|x - z\|_2. \tag{20}$$

Substituting this into the energy functional yields a tractable upper bound:

$$\mathcal{E}(\alpha, \beta) \le K^2 \int_0^1 \mathbb{E} \left[ \|\mathbf{X}_t - \mathbf{Z}_t\|^2 \right] dt. \tag{21}$$

Minimizing $\mathcal{E}$ requires minimizing this upper bound, denoted as $\mathcal{J}(\alpha, \beta) = \mathbb{E}[\|\mathbf{X}_t - \mathbf{Z}_t\|^2]$.

**Minimization of Trajectory Displacement.** We analyze the instantaneous displacement $\Delta_t = \mathbf{X}_t - \mathbf{Z}_t$ inside the expectation. Substituting the trajectory definitions:

$$\Delta_t = (\alpha_t \tilde{\mathbf{X}}_t + \beta_t \mathbf{X}_T) - ((1-t)\tilde{\mathbf{X}}_t + t\mathbf{X}_T). \tag{22}$$

Rearranging terms by basis vectors $\tilde{\mathbf{X}}_t$ and $\mathbf{X}_T$:

$$\Delta_t = (\alpha_t - (1-t))\tilde{\mathbf{X}}_t + (\beta_t - t)\mathbf{X}_T. \tag{23}$$

The squared norm is:

$$\|\Delta_t\|^2 = (\alpha_t - (1-t))^2 \|\tilde{\mathbf{X}}_t\|^2 + (\beta_t - t)^2 \|\mathbf{X}_T\|^2 + 2(\alpha_t - (1-t))(\beta_t - t)\langle \tilde{\mathbf{X}}_t, \mathbf{X}_T \rangle. \tag{24}$$

Taking the expectation as follows:

$$\begin{cases} \mathbb{E}[\cdot] : \mathbb{E}[\|\tilde{\mathbf{X}}_t\|^2] = C_1 > 0, & \text{(Finite energy of signal)}; \\ \mathbb{E}[\|\mathbf{X}_T\|^2] = d, & \text{(Expected norm of } d\text{-dimensional Gaussian noise)}; \\ \mathbb{E}[\langle \tilde{\mathbf{X}}_t, \mathbf{X}_T \rangle] = 0, & \text{(Signal and terminal noise are uncorrelated)}; \end{cases} \tag{25}$$

Thus, the objective function simplifies to a sum of non-negative quadratic terms as

$$\mathcal{J}_t(\alpha_t, \beta_t) = C_1(\alpha_t - (1-t))^2 + d(\beta_t - t)^2. \tag{26}$$

**Global Optimality**. Since $C_1 > 0$ and $d > 0$, the quadratic form $\mathcal{J}_t$ is strictly convex with respect to $\alpha_t$ and $\beta_t$. The global minimum of zero is achieved if and only if each quadratic term vanishes independently:

$$\begin{cases} (\alpha_t - (1 - t))^2 = 0 \implies \alpha_t = 1 - t; \\ (\beta_t - t)^2 = 0 \implies \beta_t = t. \end{cases} \tag{27}$$

We can finally draw the following conclusion. Any choice where $\alpha_t \neq 1 - t$ or $\beta_t \neq t$ yields $\mathcal{J}_t > 0$, strictly increasing the upper bound on the Adaptation Energy $\mathcal{E}$. Therefore, the linear schedule is the unique minimizer that preserves the pretrained signal-to-noise alignment.

### B.3 Proof of Geometric Linearity (Geodesic Trajectory Expectation)

**Proposition B.1** (Validation of Trajectory Minimal Energy). *Let $\mathbf{X}_t$ be the stochastic process defined by the Energy-oriented Geodesic Bridge SDE $\mathbf{X}_t = (1 - t)\widetilde{\mathbf{X}}_t + t\mathbf{X}_T$ from Eq. (6). The expectation of this process, $\mu(t) = \mathbb{E}[\mathbf{X}_t]$, coincides exactly with the manifold geodesic $\gamma(t)$ for all $t \in [0, T_0]$.*

*Proof.* □

**Variational Definition of the Geodesic**. Let $\mathcal{M}$ be the data manifold equipped with a Riemannian metric $g$. The geodesic $\gamma(t)$ connecting the clean distribution $\pi_0$ and the degraded distribution $\pi_{T_0}$ is the curve that minimizes the energy functional:

$$E[\gamma] = \frac{1}{2} \int_0^{T_0} \|\dot{\gamma}(t)\|^2 dt. \tag{28}$$

The critical point of this functional satisfies the Euler-Lagrange equation, known as the geodesic equation Kriz & Pultr (2013):

$$\nabla_{\dot{\gamma}} \dot{\gamma} = 0 \implies \ddot{\gamma}(t) + \Gamma(\gamma)(\dot{\gamma}, \dot{\gamma}) = 0. \tag{29}$$

**Construction of the Stochastic Process**. We construct the stochastic process $\mathbf{X}_t$ governing the bridge via the following Stochastic Differential Equation (SDE) Song et al. (2021):

$$d\mathbf{X}_t = u(\mathbf{X}_t, t)dt + \sigma d\mathbf{W}_t, \tag{30}$$

where the total drift $u(\mathbf{X}_t, t)$ is designed. The design of $u(\mathbf{X}_t, t)$ should enable the diffusion process trace the manifold geodesic trajectory. Thus, we introduce an **error feed-back control dynamics**. It starts with defining the error in the Riemannian manifold of

$$\mathbf{e}(t) = \log_{\mu(t)}(\gamma(t)). \tag{31}$$

where $\log_{\mathbf{X}_t}(\gamma(t))$ is the Riemannian logarithm map, representing the tangent vector at $\mathbf{X}_t$ pointing towards the geodesic $\gamma(t)$. To make the drift traces the geodesic trajectory, such an error term should gradually decay. We thus design its gradient as

$$\dot{\mathbf{e}}(t) = -\alpha \mathbf{e}(t), \tag{32}$$

where $\alpha > 0$ denotes a scalar factor for error feedback. By substituting the Eq. (31) in to the aforementioned equations, we have

$$\dot{\mu}(t) = \dot{\gamma}(t) + \alpha \log_{\mu(t)}(\gamma(t)). \tag{33}$$

Note that we have the following expectation drifting process as $\frac{d}{dt}\mathbb{E}[\mathbf{X}_t] = \mathbb{E}(u(\mathbf{X}_t, t))$ for Eq. (33). To ensure the requirement of Eq. (33), we can parameterize $u(\mathbf{X}_t, t)$ as

$$u(\mathbf{X}, t) = \dot{\gamma}(t) + \alpha \log_{\mu(t)}(\gamma(t)). \tag{34}$$

**Derivation of the Mean Evolution ODE**. We examine the time evolution of the first moment (mean), $\mu(t) = \mathbb{E}[\mathbf{X}_t]$. Taking the expectation of the SDE integral form (noting $\mathbb{E}[dW_t] = 0$), we have

$$\frac{d\mu(t)}{dt} = \mathbb{E}[u(\mathbf{X}_t, t)] = \mathbb{E}[b(\mathbf{X}_t)] + \alpha \mathbb{E}[\log_{\mathbf{X}_t}(\gamma(t))]. \tag{35}$$

**Linearization via Taylor Expansion**. Assuming the distribution of $\mathbf{X}_t$ is concentrated around its mean $\mu(t)$, we linearize the Riemannian logarithm map around $\mu(t)$:

$$\mathbb{E}[\log_{\mathbf{X}_t}(\gamma(t))] \approx \log_{\mu(t)}(\gamma(t)). \tag{36}$$

In a local coordinate chart where the manifold is approximately Euclidean (due to the noise), or assuming the ambient space is Euclidean (as is common in diffusion models), the logarithm map simplifies to the vector difference:

$$\log_{\mu(t)}(\gamma(t)) \approx \gamma(t) - \mu(t). \tag{37}$$

Furthermore, the transport velocity of the mean must account for the target's motion. The effective drift driving the mean tracking error dynamics is relative to the geodesic velocity $\dot{\gamma}(t)$. Thus, the evolution equation becomes

$$\frac{d\mu(t)}{dt} = \dot{\gamma}(t) + \alpha\left[\gamma(t) - \mu(t)\right]. \tag{38}$$

**Analysis of Error Dynamics**. Define the deviation from the geodesic as $\delta(t) = \mu(t) - \gamma(t)$. Differentiating with respect to time:

$$\dot{\delta}(t) = \dot{\mu}(t) - \dot{\gamma}(t), \tag{39}$$

Substituting the mean evolution equation derived in Linearization via Taylor Expansion:

$$\dot{\delta}(t) = [\dot{\gamma}(t) + \alpha(\gamma(t) - \mu(t))] - \dot{\gamma}(t), \tag{40}$$

$$\dot{\delta}(t) = -\alpha(\mu(t) - \gamma(t)), \tag{41}$$

$$\dot{\delta}(t) = -\alpha\delta(t). \tag{42}$$

This is a homogeneous first-order linear ordinary differential equation. The general solution is

$$\delta(t) = \delta(\epsilon)e^{-\alpha t}. \tag{43}$$

The Energy-oriented Geodesic bridge is initialized exactly on the geodesic at $t = T$ for $\mathbb{E}(\mathbf{X}_T) = \mathbf{Y}$,

$$\lim_{\epsilon \to T} \delta(\epsilon) = 0. \tag{44}$$

Therefore, the initial deviation is $\delta(0) = 0$. Then, we have

$$\delta(t) = 0 \cdot e^{-\alpha t} = 0 \quad \forall t, \tag{45}$$

$$\mu(t) = \gamma(t). \tag{46}$$

Thus, the mean trajectory of the process is identical to the geodesic $\gamma(t)$ by construction.

### B.4 PROOF OF ENERGETIC OPTIMALITY (ZERO CONTROL ENERGY)

**Proposition B.2.** *(**Energetic Optimality of the Geodesic Bridge**) Let $\mathcal{P}_{E-Bridge}$ be the probability measure of the Energy-oriented diffusion Bridge (E-Bridge) process $\mathbf{X}_t$ defined on the interval $[0, T_0]$. The kinetic energy cost $E_K[\mu]$ required to steer the mean trajectory $\mu(t) = \mathbb{E}[\mathbf{X}_t]$ along the manifold geodesic $\gamma(t)$ is identically zero.*

**Definition of Control Energy (Onsager-Machlup Functional)**. From the perspective of Stochastic Optimal Control (SOC), the energy cost of a diffusion process is quantified by the kinetic term of the Onsager-Machlup action functional Dürr & Bach (1978); Raja et al. (2025). This function measures the deviation of the process's mean velocity from its expected instantaneous drift.

For a process with mean path $\mu(t)$, the trajectory energy is defined as:

$$E_K[\mu] = \frac{1}{2}\int_0^{T_0} \|\dot{\mu}(t) - \mathbb{E}_{\mathbf{X}_t \sim p_t}[b_{\text{total}}(t, \mathbf{X}_t)]\|_{\mu(t)}^2\, dt, \tag{47}$$

where $\dot{\mu}(t)$ is the actual velocity of the mean trajectory; $b_{\text{total}}(t, \mathbf{X}_t)$ is the total drift vector field governing the Stochastic Differential Equation (SDE); and $\| \cdot \|_{\mu(t)}$ is the Riemannian metric norm at the point $\mu(t)$.

**Dynamics of the Geodesic Bridge**. The geodesic bridge process is governed by the following SDE on the manifold:

$$d\mathbf{X}_t = \underbrace{(b(\mathbf{X}_t) + b_{geo}(t, \mathbf{X}_t))}_{b_{\text{total}}(t, \mathbf{X}_t)} dt + \sigma(\mathbf{X}_t)d\mathbf{W}_t. \tag{48}$$

The specific control drift $b_{geo}$ is constructed to enforce tracing geodesic trajectories:

$$b_{geo}(t, \mathbf{X}_t) = \alpha \cdot \log_{\mathbf{X}_t}(\gamma(t)) + \mathcal{T}_{\gamma(t) \to \mathbf{X}_t}(\dot{\gamma}(t)). \tag{49}$$

The second term is the parallel transport of the geodesic velocity, often implicit in the Euclidean formulation.

**Evolution of the Expected Drift**. We calculate the expected total drift at time $t$. Substituting the definition of $b_{\text{total}}$:

$$\mathbb{E}[b_{\text{total}}(t, \mathbf{X}_t)] = \mathbb{E}[b(\mathbf{X}_t)] + \alpha\mathbb{E}[\log_{\mathbf{X}_t}(\gamma(t))] + \mathbb{E}[\mathcal{T}_{\gamma(t) \to \mathbf{X}_t}(\dot{\gamma}(t))]. \tag{50}$$

By **proposition** 4.1 (Geodesic Consistency), we have proven that the trajectory expectation coincides with the geodesic, i.e., $\mu(t) = \gamma(t)$. At the mean $\mu(t) = \gamma(t)$, the logarithmic map (error term) vanishes: $\mathbb{E}[\log_{\mathbf{X}_t}(\gamma(t))] \approx \log_{\gamma(t)}(\gamma(t)) = 0$. The transport term at the mean becomes the geodesic velocity itself: $\dot{\gamma}(t)$. Assuming the base drift $b(\mathbf{X}_t)$ is zero (standard Brownian motion) or symmetric around the geodesic, it does not contribute to the mean transport velocity.

Thus, the expected total drift is exactly the geodesic velocity:

$$\mathbb{E}[b_{\text{total}}(t, \mathbf{X}_t)] = \dot{\gamma}(t), \tag{51}$$

**Evaluation of the Energy Functional**. We substitute the actual velocity of the mean ($\dot{\mu}(t) = \dot{\gamma}(t)$) and the expected drift derived above into the energy functional equation:

$$E_K[\mu] = \frac{1}{2} \int_0^{T_0} \left\| \dot{\gamma}(t) - \underbrace{\mathbb{E}[b_{\text{total}}(t, \mathbf{X}_t)]}_{\dot{\gamma}(t)} \right\|_{\gamma(t)}^2 dt, \tag{52}$$

$$E_K[\mu] = \frac{1}{2} \int_0^{T_0} \|\dot{\gamma}(t) - \dot{\gamma}(t)\|^2 dt, \tag{53}$$

$$E_K[\mu] = \frac{1}{2} \int_0^{T_0} 0 \cdot dt = 0. \tag{54}$$

Then, we can conclude that

$$E_K[\mu_{E-Bridge}] = 0. \tag{55}$$

This formally proves that the proposed E-Bridge requires **zero net control energy** to steer the mean of the distribution along the restoration path.

In contrast, for a standard bridge (e.g., Doob's h-transform) constrained to hit a fixed endpoint $\mathbf{Y}$, the drift is $b_{add} = \nabla \log h(t, x)$. The expected drift $\mathbb{E}[b_{add}]$ generally does not equal the velocity of the mean path $\dot{\mu}(t)$ except in the trivial case of zero curvature and no constraints. The "forcing" required to satisfy the boundary condition $\mathbf{X}_1 = \mathbf{Y}$ creates a non-zero residual $\|\dot{\mu} - \mathbb{E}[b]\| > 0$, resulting in strictly positive energy $E_K > 0$. We prove it in the next section.

## C GCB V.S. ORNSTEIN–UHLENBECK PROCESS: OPTIMALITY UNDER CONSTRAINT

**Expectation Integral**. We compare the two processes based on the objective: Move the distribution mean $\mu(t)$ from the clean image $\mathbf{X}_0$ to the degraded image $\mathbf{Y}$ over time $t \in [0, 1]$.

**Proposed E-Bridge (Geodesic)**: the mean trajectory is a **Linear Interpolation** (constant velocity). We can simplify the trajectory of the expectation of the state as

$$\mu_{E-Bridge}(t) = (1-t)\mathbf{X}_0 + t\mathbf{Y}. \tag{56}$$

While the corresponding velocity can be calculated as

$$v_{E-Bridge}(t) = \mathbf{Y} - \mathbf{X}_0. \tag{57}$$

Note that the velocity is constant. Moreover, for the general OU Process (Mean-Reverting). The mean trajectory is an **Exponential Decay** (variable velocity) towards $\mathbf{Y}$.

$$\mu_{OU}(t) = \mathbf{Y} + (\mathbf{X}_0 - \mathbf{Y})e^{-\theta t}. \tag{58}$$

The velocity is

$$v_{OU}(t) = -\theta(\mathbf{X}_0 - \mathbf{Y})e^{-\theta t}. \tag{59}$$

Then, transportation energy is the integral of squared velocity:

$$E = \int_0^1 \|v(t)\|^2 dt \tag{60}$$

Let's calculate the energy required to cover a specific displacement $D = \|\mu(1) - \mu(0)\|$. By Jensen's Inequality McShane (1937) (applied to the convex function $f(x) = x^2$), for any path satisfying the displacement $D$:

$$\underbrace{\int_0^1 \|v(t)\|^2 dt}_{\text{Mean Squared Speed}} \geq \underbrace{\left(\int_0^1 \|v(t)\| dt\right)^2}_{\text{Squared Mean Speed}} \geq \|D\|^2. \tag{61}$$

**Equality holds if and only if the velocity $v(t)$ is constant.**

Due to the **E-Bridge** Has constant velocity. Thus, it achieves the **theoretical minimum energy** for the distance it covers.

$$E_{E-Bridge} = \|\mathbf{Y} - \mathbf{X}_0\|^2. \tag{62}$$

However, **General OU** has variable velocity (starts fast, slows down). Therefore, for the same displacement, its energy is strictly higher.

$$E_{OU} > E_{E-Bridge}. \tag{63}$$

## D  ANALYSIS OF SCHRÖDINGER DIFFUSION BRIDGE

The further mathematical analysis of the Schrödinger Diffusion Bridge, framed as a continuation of the previous analysis of Doob's $h$-bridge, which proves its inherent drawbacks in terms of computational complexity and energetic inefficiency.

### Analysis of the Schrödinger Diffusion Bridge

The Schrödinger Bridge (SB) problem offers a powerful and flexible framework for connecting two probability distributions, representing a significant conceptual advance over the standard Doob's h-transform. It is deeply connected to the theory of entropy-regularized optimal transport (EOT). The SB finds the stochastic process that transports an initial distribution $\pi_0$ to a final distribution $\pi_T$ while being "closest" in a probabilistic sense—specifically, by minimizing the Kullback-Leibler (KL) divergence—to a reference process, such as Brownian motion. While this provides an elegant theoretical solution, we will prove that this entropic optimality comes at the cost of significant computational complexity and, like the Doob's bridge, results in a trajectory that is not energetically optimal in the sense of the Onsager-Machlup action.

### Mathematical Formulation of the Schrödinger Bridge Problem

Let $\Pi_{ref}$ be the trajectory measure of a reference diffusion process on a manifold $M$ (e.g., Brownian motion) over the time interval. The Schrödinger Bridge problem seeks to find an optimal trajectory measure $\Pi^\star$ that solves the following variational problem:

$$\Pi^\star = \arg\min_{\Pi} \{KL(\Pi|\Pi_{ref}) \quad \text{s.t.} \quad \Pi_0 = \pi_0, \quad \Pi_T = \pi_T\}. \tag{64}$$

Here, $\mathrm{KL}(\Pi|\Pi_{ref})$ is the Kullback-Leibler divergence, and $\Pi_0$ and $\Pi_T$ are the marginal distributions of the trajectory measure $\Pi$ at times $t = 0$ and $t = T$, respectively. This formulation seeks the "most likely" random evolution connecting the two distributions, given the reference dynamics.

The solution to this problem, $\Pi^\star$, is a Markovian diffusion process described by a pair of forward and backward stochastic differential equations (SDEs). For a reference process $d\mathbf{X}_t = \sigma d\mathbf{W}_t$, the optimal forward process $(\mathbf{X}_t)$ and backward process $(\mathbf{Y}_t)$ are given by a coupled system of SDEs:

$$d\mathbf{X}_t = b_F(t, \mathbf{X}_t)dt + \sigma d\mathbf{W}_t, \quad \mathbf{X}_0 \sim \pi_0, \tag{65}$$

$$d\mathbf{Y}_t = b_B(t, \mathbf{Y}_t)dt + \sigma d\bar{\mathbf{W}}_t, \quad \mathbf{Y}_T \sim \pi_T. \tag{66}$$

The drifts $b_F$ and $b_B$ are learned to ensure that the marginal distribution of $\mathbf{X}_t$ evolves to $\pi_T$ at time $T$, and the marginal of $\mathbf{Y}_t$ (evolving backward in time from $T$) matches $\pi_0$ at time 0.

**Proof of Drawback 1: High Computational Cost via Iterative Proportional Fitting (IPF)**

The primary method for numerically solving the Schrödinger Bridge problem is the Iterative Proportional Fitting (IPF) procedure, a continuous-space analogue of the Sinkhorn algorithm. This iterative nature is the source of the method's significant computational burden.

The IPF algorithm generates a sequence of trajectory measures $(\Pi_n)_{n\in\mathbb{N}}$ that alternate between satisfying the initial and terminal marginal constraints. Starting with the reference measure $\Pi_0 = \Pi_{ref}$, the iterative steps are defined as:

- **Backward Step**: Find $\Pi_{2n+1} = \arg\min_\Pi\{\mathrm{KL}(\Pi|\Pi_{2n}) \text{ s.t. } \Pi_T = \pi_T\}$. This step adjusts the process to match the target distribution $\pi_T$.
- **Forward Step**: Find $\Pi_{2n+2} = \arg\min_\Pi\{\mathrm{KL}(\Pi|\Pi_{2n+1}) \text{ s.t. } \Pi_0 = \pi_0\}$. This step adjusts the new process to match the initial distribution $\pi_0$.

In modern implementations like Diffusion Schrödinger Bridge (DSB), each of these steps requires training a neural network to approximate the drift of the corresponding time-reversed SDE This leads to several computational drawbacks:

- **Dual Training**: Unlike standard diffusion models that only learn the backward (generative) process, the SB framework requires training both a forward and a backward process drift function at each iteration of the IPF algorithm. This immediately doubles the training load per iteration;
- **Iterative Refinement**: The IPF procedure is iterative. The entire process of simulating trajectories and training both forward and backward drift networks must be repeated multiple times until the marginals converge. This multiplicative effect makes the total computational cost much higher than that of a single-pass diffusion model; and
- **Error Accumulation**: Numerical methods for approximating SBs, particularly those based on IPF, are known to be susceptible to the accumulation of errors across iterations, which can affect the stability and accuracy of the final solution.

Therefore, while the SB framework is more flexible than a simple noising process, this flexibility is achieved at a substantially higher computational cost, which is a significant drawback for practical applications.

**Proof of Drawback 2: Energetic Sub-optimality**

The Schrödinger Bridge is optimal in an entropic sense (minimizing KL divergence), but this does not imply optimality in an energetic sense (minimizing the Onsager-Machlup kinetic action). The trajectory taken by the mean of the SB process is an "entropic interpolation" between the two distributions, not a geometric one.

The objective of the SB is to find the most probable random evolution, not the most efficient deterministic path. The resulting process is a complex balance between the underlying diffusive tendency of the reference process and the control drifts required to meet the boundary conditions. The two variational problems, minimizing kinetic action (as in the Energy-oriented Geodesic Bridge) and minimizing KL divergence (as in the SB), are fundamentally different. Their relationship is characterized by a Fisher information functional, which quantifies the difference between the two objectives.

Let $\mu_{SB}(t)$ be the trajectory expectation of the Schrödinger Bridge process. The total drift of the forward process is $b_F(t, z)$. The trajectory energy of the mean is given by:

$$E_K = \frac{1}{2} \int_0^T ||\dot{\mu}_{SB}(t) - E[b_{ref}(\mathbf{X}_s)]_{s=t}||^2_{\mu_{SB}(t)} dt, \tag{67}$$

where $b_{ref}$ is the drift of the reference process (e.g., zero for Brownian motion in Euclidean space, or the geometric drift on a manifold). Since $\dot{\mu}_{SB}(t) = E[b_F(t, \mathbf{X}_s)]_{s=t}$, the energy is:

$$E_K \approx \frac{1}{2} \int_0^T ||E[b_F(t, \mathbf{X}_s) - b_{ref}(\mathbf{X}_s)]_{s=t}||^2_{\mu_{SB}(t)} dt. \tag{68}$$

The control drift, $b_F - b_{ref}$, is learned through the iterative IPF procedure to satisfy the terminal condition $\Pi_T = \pi_T$. This drift is generally non-zero and is not designed to follow a geodesic. It steers the mean along a trajectory dictated by entropic considerations. Consequently, the trajectory energy $E_K$ is manifestly greater than zero.

While the SB is more structured than the Doob's h-transform bridge (as it learns both forward and backward processes), it does not achieve the zero-energy mean transport of the Energy-oriented Geodesic Bridge. The control energy is expended to steer the mean along a trajectory that is entropically optimal but geometrically and energetically suboptimal. This proves that the Schrödinger Bridge, despite its theoretical elegance, still represents a high-cost solution compared to a geometrically-derived bridge.

## E ANALYSIS OF E-BRIDGE WITHIN THE STOCHASTIC INTERPOLANT FRAMEWORK

**General Stochastic Interpolant Framework**

The general Stochastic Interpolant (SI) framework Albergo et al. (2023) defines a time-dependent process $\mathbf{X}_t$ that interpolates between two distributions $\rho_0$ and $\rho_1$ using a stochastic map. The interpolant is generally defined as:

$$\mathbf{X}_t = \alpha(t)\mathbf{X}_0 + \beta(t)\mathbf{X}_1 + \gamma(t)\mathbf{Z}, \quad t \in [0, 1], \tag{69}$$

where $\mathbf{X}_0 \sim \rho_0$ represents source distribution, $\mathbf{X}_1 \sim \rho_1$ is the target distribution. $\mathbf{Z} \sim \mathcal{N}(0, I)$ is an auxiliary Gaussian variable. $\alpha(t), \beta(t)$ are smooth interpolation coefficients satisfying $\alpha(0) = 1, \alpha(1) = 0$ and $\beta(0) = 0, \beta(1) = 1$. $\gamma(t)$ controls the noise level, typically with $\gamma(0) = \gamma(1) = 0$ for bridge processes.

The objective of SI methods is to learn a velocity field $v_t(x)$ (and potentially a score field $s_t(x)$) that generates the probability flow ODE:

$$\frac{d\mathbf{X}_t}{dt} = v_t(\mathbf{X}_t). \tag{70}$$

The optimal velocity field $v_t^*(x)$ is defined as the conditional expectation of the time derivative of the interpolant:

$$v_t^*(x) = \mathbb{E}[\dot{\mathbf{X}}_t | \mathbf{X}_t = x]. \tag{71}$$

**Brownian Bridge as a Specific Instance of SI**

The standard Brownian Bridge (BB) is a specific instance of the SI framework. By selecting the following coefficients:

$$\alpha(t) = 1 - t, \quad \beta(t) = t, \quad \gamma(t) = \sigma\sqrt{t(1-t)} \tag{72}$$

The interpolant becomes:

$$\mathbf{X}_t^{BB} = (1-t)\mathbf{X}_0 + t\mathbf{X}_1 + \sigma\sqrt{t(1-t)}\mathbf{Z}. \tag{73}$$

Taking the time derivative $\dot{\mathbf{X}}_t^{BB}$:

$$\dot{\mathbf{X}}_t^{BB} = (\mathbf{X}_1 - \mathbf{X}_0) + \frac{\sigma(1-2t)}{2\sqrt{t(1-t)}}\mathbf{Z}. \tag{74}$$

To find the velocity field $v_t^{BB}(x)$, we express $\mathbf{Z}$ in terms of $\mathbf{X}_t$ using the interpolant equation:

$$\mathbf{Z} = \frac{\mathbf{X}_t - (1-t)\mathbf{X}_0 - t\mathbf{X}_1}{\sigma\sqrt{t(1-t)}}. \tag{75}$$

Substituting $\mathbf{Z}$ into $\dot{\mathbf{X}}_t^{BB}$ and taking the expectation $\mathbb{E}[\cdot|\mathbf{X}_t = x]$, we obtain the vector field for the standard Brownian Bridge:

$$v_t^{BB}(x) = \frac{\mathbb{E}[\mathbf{X}_1|\mathbf{X}_t = x] - x}{1-t} + \underbrace{\frac{x - \mathbb{E}[\mathbf{X}_0|\mathbf{X}_t = x]}{t}}_{\text{if } \rho_1 \text{ is noise, this term varies}}. \tag{76}$$

Specifically, focusing on the bridge structure connecting data to data (or data to noise), the dominant term governing the dynamics as $t \to 1$ is:

$$v_t^{BB}(x) \approx \frac{\mathbf{X}_1 - x}{1-t}. \tag{77}$$

**Theoretical Limitations of Standard SI (Brownian Bridge)**

Although mathematically elegant, the standard BB formulation within SI suffers from two critical issues at the boundary $t = 1$:

**A. Singularity of the Vector Field**   As $t \to 1$, the denominator $(1-t) \to 0$. Unless the numerator $(\mathbf{X}_1 - x)$ converges to 0 faster than the denominator (which is not guaranteed during training, especially with neural network approximation errors), the velocity field diverges:

$$\lim_{t \to 1} \|v_t^{BB}(x)\| \to \infty. \tag{78}$$

This singularity leads to: **1) Unstable Training:** The regression target for the neural network becomes unbounded. **2) Numerical Stiffness:** The ODE solver requires extremely small steps near $t = 1$, increasing inference cost.

**B. Unbounded Lipschitz Constant**   The difficulty of learning the flow is characterized by the Lipschitz constant $L_t$ of the velocity field $v_t(x)$. For the Brownian Bridge:

$$\nabla_x v_t^{BB}(x) \propto -\frac{1}{1-t}I. \tag{79}$$

The Lipschitz constant $L_t \approx \frac{1}{1-t}$ is not integrable over $[0, 1]$:

$$\int_0^1 L_t dt = \int_0^1 \frac{1}{1-t} dt = \infty. \tag{80}$$

This implies that the flow becomes infinitely sensitive to perturbations as $t \to 1$, making the learning problem ill-posed at the boundary.

**E-Bridge: Regularization via Horizon Truncation ($T_0$)**

Our proposed E-Bridge method solves the limitation of Brownian Bridge by introducing a **truncated horizon** $T_0 < 1$. The process is defined on $t \in [0, T_0]$.

**A. Bounded Vector Field**   By restricting $t \le T_0$, the denominator in the velocity field is strictly bounded from below by $1 - T_0 > 0$. The maximum norm of the velocity field is bounded:

$$\sup_{t \in [0,T_0]} \|v_t^{E-Bridge}(x)\| \le C \cdot \frac{1}{1 - T_0} < \infty, \tag{81}$$

where $C$ depends on the bounded domain of the data. This eliminates the singularity.

**B. Finite Lipschitz Constant**   The Lipschitz constant for the E-Bridge flow is bounded:

$$L_{E-Bridge} = \sup_{t \in [0,T_0]} \left\| \nabla_x v_t^{E-Bridge}(x) \right\| \approx \frac{1}{1 - T_0}. \tag{82}$$

Consequently, the total accumulation of the Lipschitz constant (which relates to the error bound of the ODE solver) is finite:

$$\int_0^{T_0} L_t dt \approx \int_0^{T_0} \frac{1}{1-t} dt = \ln\left(\frac{1}{1 - T_0}\right) < \infty. \tag{83}$$

## F  ADDITIONAL EXPERIMENTAL RESULTS

### F.1  COMPUTATIONAL EFFICIENCY COMPARISON

Table F-5 highlights our method's exceptional efficiency: even with the largest model parameters, it incurs the lowest total inference time by requiring significantly fewer NFEs, significantly outperforming other state-of-the-art approaches.

### F.2  REAL-WORLD IMAGE RESTORATION

We provide qualitative results on real-world super-resolution. We evaluated our E-Bridge model on real-world low-quality images from Ji et al. (2020). Note that the model has never seen these specific real-world degradation distributions during training. As illustrated in Fig. F-3, our E-Bridge demonstrates robust generalization capabilities. This experiment empirically validates that the dependence on paired data is not a hindrance to practicality. By leveraging robust synthetic data pipelines, the proposed E-Bridge can also achieve high-performance real-world restoration without requiring test-time optimization or degradation estimation, offering a superior trade-off between offline training cost and online inference quality.

Table F-5: Computational efficiency comparison on the $1024 \times 1024$ image denoising.

| Method | Training Parameters | Inference Time per Step | NFE | Total Inference Time |
|---|---|---|---|---|
| AutoDIR | 115.9 M | 0.823s | 25 | 20.58s |
| UniDB | 137.1 M | 1.243s | 100 | 124.30s |
| IRBridge | 361.3 M | 0.263s | 100 | 26.30s |
| UniDB++ | 137.1 M | 1.224s | 10 | 12.24s |
| Ours | 743.8 M | 0.673s | 10 | 6.73s |

## G  LIMITATION AND FUTURE WORK

Although our method demonstrates superior perceptual performance and inference efficiency, several limitations remain that guide our future research directions.

### G.1  LIMITATION

**Spatial Heterogeneity Conflict:** As observed in tasks like raindrop removal, our method achieves SOTA perceptual scores but can be numerically inferior in reference-based metrics (e.g., PSNR). This is a direct consequence of using a global scalar parameter $T_0$. A uniform $T_0$ enforces a single trade-off across the entire image, making it difficult to simultaneously optimize for generative power (required to inpaint occluded regions like raindrops) and information preservation (required to maintain fidelity in clean background regions).

**Computational Overhead:** Despite significantly reducing the Number of Function Evaluations (NFE) to 1-10 steps, the reliance on the massive Flux-dev backbone necessitates substantial VRAM. This computational footprint currently hinders deployment on consumer-grade hardware or edge devices compared to other baselines.

### G.2  FUTURE WORK

**Spatially Adaptive $T_0$:** To resolve the heterogeneity conflict, we plan to extend the E-Bridge framework from a global scalar to a Spatially Adaptive Map $T_0(h, w)$. This will allow the model to dynamically apply high generative strength locally to degraded regions (e.g., raindrops) while enforcing high fidelity in clean areas, optimizing the trade-off pixel-by-pixel.

**Model Compression:** We plan to compress the generative capabilities of the large-scale Flux-dev backbone into a lightweight network. This process could employ trajectory alignment to minimize perceptual discrepancies between the student's single-step predictions and the teacher's geodesic mapping, feature-level distillation to align intermediate semantic representations, and adversarial objectives to ensure the compact model retains the sharp, photorealistic textures of the original E-Bridge without over-smoothing.

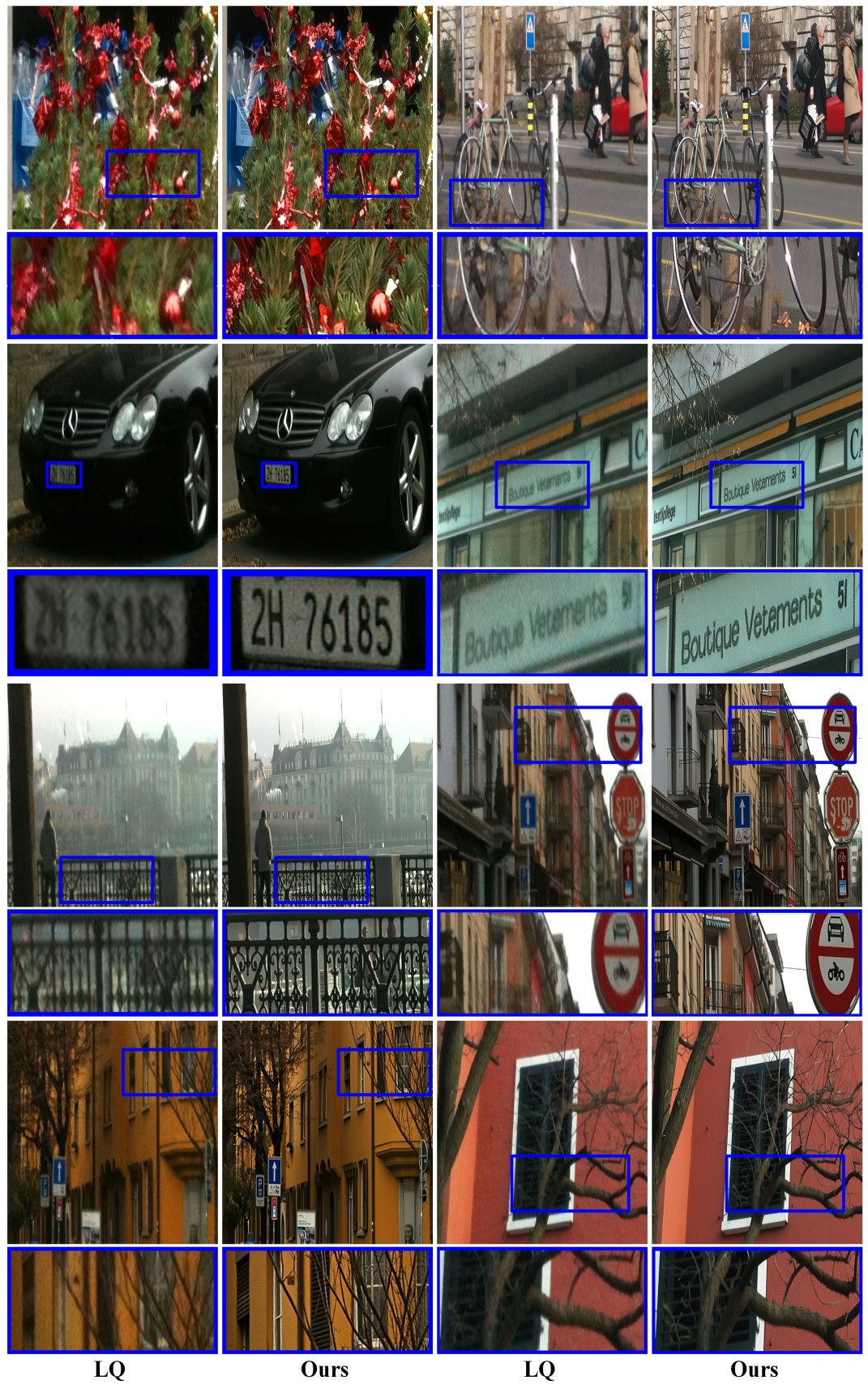

**LQ**        **Ours**        **LQ**        **Ours**

Figure F-3: Additional visual results on image super-resolution in the wild.

