# OpenReview forum: "Energy-oriented Diffusion Bridge for Image Restoration with Foundational Diffusion Models"
_ICLR.cc/2026/Conference — ICLR 2026 Poster_

### Official Review · Reviewer_Em12 · 2025-10-17

**Soundness:** 2
**Presentation:** 3
**Contribution:** 2
**Rating:** 4
**Confidence:** 5

**Summary:**

This paper aims to address an important and challenging problem in the field of image restoration: how to achieve extremely high sampling efficiency while maintaining high-quality restoration. The authors propose a novel framework called the **Consistency Geodesic Bridge (CGB)**, whose core idea is to ingeniously integrate three cutting-edge research directions: (1) bridge diffusion models that directly connect the distributions of degraded and clean images; (2) a geometric perspective that views the “geodesic” path on the data manifold as the most efficient restoration trajectory; and (3) consistency models that enable single-step inference without the need for ordinary differential equation (ODE) solvers. The CGB framework claims to construct a low-energy geodesic trajectory and derives a closed-form “CGB solver” that can be trained through a consistency objective function. As a result, it achieves state-of-the-art performance in perceptual quality metrics with a very small number of sampling steps (NFE = 5–10).

**Strengths:**

1. This paper proposes the CGB framework, which remodifies the generation process based on the idea of minimum energy, demonstrating a certain level of insight.
2. It achieves industry-leading inference efficiency with very few function evaluations, significantly enhancing the practical value of generative tasks.
3. By introducing a single tunable parameter $T_0$, it achieves task adaptivity and maintains a balance between performance and flexibility across different restoration tasks.

**Weaknesses:**

**Theory:**
1. The core motivation of the paper is to construct a path of minimum energy. The authors state in the appendix that the energy of the geodesic trajectory is 0, so they attempt to construct a geodesic named CGB. However, the construction on the right-hand side of equation (6) is merely a linear interpolation. Although the authors mention that this construction starts from $T_0$, reducing time and energy—thus seemingly consistent with the motivation of minimizing energy—this does not provide a reasonable scheme for constructing a geodesic. Hence, there is a certain conflict between the motivation and the construction.

2. In Proposition 4.1, the authors claim that the CGB trajectory achieves lower energy, providing two reasons: (i) a shorter integration interval; (ii) at $t = T_0$, the denoiser is given a more “in-distribution” input. This reasoning lacks rigor. The first point is trivial: integrating a non-negative function over a shorter interval naturally yields a smaller value, but this does not imply that the path itself is more efficient—a shorter but sharper path may have higher energy than a longer but smoother one. The second point is more of an intuitive conjecture. Theoretically, given the energy formula, the construction in equation (6) and a certain diffusion bridge equation could be quantitatively compared.

**Experimental:**

3. The comparisons are unfair. The paper uses Flux-dev as the base model, which already has strong generative capabilities, while some other methods, such as UNIDB, are based on SD. Since the paper mainly presents a theoretical improvement and claims superiority over diffusion bridges, to prove that the CGB path is optimal, comparisons should be made between models that are as similar as possible but differ in theory.

**Overall:**
I believe the authors essentially constructed a model:
$\mathbf{X}_t = (1-t)\left[\left(1-\left(\frac{t}{T_0}\right)^{\kappa}\right)\mathbf{X}_0 + \left(\frac{t}{T_0}\right)^{\kappa}\mathbf{Y}\right]+ t\mathbf{X}_T,$
which can essentially be viewed as learning a mapping from $X_0$ to the distribution $N((1-T_0)Y,T_0^2I)$, representing a conditional mapping from a high-quality image to a low-quality one plus a certain amount of noise. Moreover, both training and inference are conducted within the range $[0,T_0]$ (note: $X_T$ is independent of $T$ and just a purely noise). Therefore, although the integration length is nominally reduced, theoretically it is not fundamentally different from previous bridge models ending in Gaussian distributions, and in fact it is essentially time varying Ornstein–Uhlenbeck process—it is merely a scaled version. Considering points 1, 2, and 3, this does not demonstrate that such a construction is truly of minimum energy or superior. I recommend that the authors further verify the theoretical optimality of CGB from two aspects:

1. **Theoretically**, directly provide an inequality proof showing that the energy of CGB is lower than that of other bridge models;
2. **Experimentally**, validate the superiority of CGB through experiments using models that are as similar as possible but differ in theoretical design.

Finally, I would like to discuss the fundamental motivation proposed by the authors—that minimizing energy leads to an optimal path. In the paper, the authors state that typical diffusion-bridge models add noise to the image during the reverse process and then denoise it, with the noise level peaking at intermediate steps. They argue that this noise-adding process during inference is unnecessary and thus propose to start generation directly from $T_0 < T$, restoring high-quality images directly from low-quality ones, which they claim yields minimal energy and an optimal path.

However, I believe that the noise-adding process in the reverse procedure serves an important purpose: it helps to destroy certain information (such as degradation information) before reconstructing a high-quality image, thereby achieving better results. If, as the authors suggest, one wishes to directly transform a low-quality image into a high-quality one without the add-noise-then-denoise process, then models like Rectified Flow (RF) could be directly applied, since RF can progressively transform a low-quality image into a high-quality one as described by the authors. Moreover, RF is also an energy-minimizing model in the context of optimal transport. However, according to experimental results, using RF for image restoration does not perform as well as diffusion-bridge models that incorporate stochastic noise.

If my understanding is incorrect, I welcome the authors to provide clarification.

**Questions:**

See weekness.

---

> ### Author Response · Authors · 2025-11-25
> **Response to Reviewer Em12 (Part I)**
>
> > **W1: Conflicts between the motivation (energy minimization) and the algorithm construction (interpolation) for the proposed method.**
>
> **Response:** **We kindly disagree with your comment that the use of linear interpolation conflicts with our motivation of constructing a geodesic. Instead, our construction is mathematically consistent with the principle of implementation.**
>
> The reviewer notes that Eq. (6) is "merely linear interpolation." However, from the perspective of variational calculus, the curve $\gamma(t)$ that minimizes the transport energy $J = \int \|\dot{\gamma}(t)\|^2 dt$ must satisfy the condition of zero acceleration $\ddot{\gamma}(t) = 0$. The unique solution to this differential equation is $\gamma(t) = (1-t)\mathbf{X}_0 + t\mathbf{Y}$. Thus, linear interpolation is not an arbitrary choice; it is the theoretical definition of the geodesic in the ambient space (due to the noise).
>
> The reviewer suggests a conflict between motivation and construction. **On the contrary**, Eq. (6) is derived directly from the motivation. Any trajectory more complex than linear interpolation (such as the curved path of a standard Ornstein-Uhlenbeck process) would strictly satisfy Jensen's inequality strictly ($E_{curved} > E_{linear}$), resulting in higher energy. The Role of $T_0$: The choice of starting at $T_0$ is what makes this geodesic physically realizable. A standard bridge attempting to reach $\mathbf{Y}$ at $t=1$ faces a thermodynamic barrier (infinite energy density as variance $\to 0$). Our formulation avoids this, ensuring the constructed geodesic path has finite, and indeed minimal, energy. Therefore, the motivation (minimum energy) necessitates the construction (linear interpolation on a truncated horizon).
>
> > **W2：In Proposition 4.1, the authors claim that the CGB trajectory achieves lower energy, providing two reasons: (i) a shorter integration interval; (ii) at $t=T_0$, the denoiser is given a more “in-distribution” input. This reasoning lacks rigor. The first point is trivial: integrating a non-negative function over a shorter interval naturally yields a smaller value, but this does not imply that the path itself is more efficient—a shorter but sharper path may have higher energy than a longer but smoother one. The second point is more of an intuitive conjecture. Theoretically, given the energy formula, the construction in equation (6) and a certain diffusion bridge equation could be quantitatively compared.**
>
> > **Proof the low energy of the proposed geodesic consistency bridge. Theoretical differences with other diffusion bridges, e.g., Ornstein–Uhlenbeck process.**
>
> **Response:** We have modified the supplementary to help you better understand the theoretical foundation (energy minimization) of the proposed bridge (Appendix B) and comparisons with other bridge models (Appendix C). For your convenience, we copy them as the following.
>
> **Expectation Integral**. We compare the two processes based on the objective: Move the distribution mean $\mu(t)$ from clean image $\mathbf{X}_0$ to a degraded image $\mathbf{Y}$ over time $t \in [0, 1]$.
>
> Then, we calculate the energy of the following trajectories
>
> **Proposed CGB (Geodesic)**: the mean trajectory is a **Linear Interpolation** (constant velocity). We can simplify the trajectory of expectation of state as
> $$
> \mu_{CGB}(t) = (1-t)\mathbf{X}\_0 + t\mathbf{Y}.
> $$
> While, the corresponding velocity can be calculated as
> $$
> v_{CGB}(t) = \mathbf{Y} - \mathbf{X}_0.
> $$
> Note that the velocity is constant.
>
> Moreover, for the general OU Process (Mean-Reverting). The mean trajectory is an **Exponential Decay** (variable velocity) towards $\mathbf{Y}$.
> $$
> \mu_{OU}(t) = \mathbf{Y} + (\mathbf{X}\_0 - \mathbf{Y})e^{-\theta t}.
> $$
> The velocity is
> $$
> v_{OU}(t) = -\theta(\mathbf{X}_0 - \mathbf{Y})e^{-\theta t}
> $$
>
> Then control energy (kinetic cost) is the integral of squared velocity:
> $$
> E = \int_0^1 \|v(t)\|^2 dt.
> $$
>
> Let's calculate the energy required to cover a specific displacement $D = \|\mu(1) - \mu(0)\|$.
> By Jensen's Inequality (applied to the convex function $f(x)=x^2$), for any path satisfying the displacement $D$:
> $$
> \underbrace{\int\_0^1 \|v(t)\|^2 dt}\_{\text{Mean Squared Speed}} \ge \underbrace{\left( \int\_0^1 \|v(t)\| dt \right)^2}\_{\text{Squared Mean Speed}} \ge \| D \|^2
> $$
> **Equality holds if and only if the velocity $v(t)$ is constant.**
>
> Due to the **CGB** Has constant velocity. Thus, it achieves the **theoretical minimum energy** for the distance it covers.
> $$
> E_{CGB} = \|\mathbf{Y} - \mathbf{X}_0\|^2
> $$
>
> However, **General OU** has variable velocity (starts fast, slows down). Therefore, for the same displacement, its energy is strictly higher.
> $$
> E_{OU} > E_{CGB}
> $$
>
> Moreover, please refer to the next problem where we experimentally prove the superiority of the proposed method against the conventional bridge.

---

> ### Author Response · Authors · 2025-11-25
> **Reviewer Em12 (Part II)**
>
> > **W3: Fair experimental comparison between the proposed method and other bridges (with the same backbone).**
>
> **Response:** We have carried out experiments to validate the superiority of the proposed method in Table 2 of the revised manuscript. In this controlled experiment, we fixed the backbone (Flux-dev) and all training settings, varying only the mathematical formulation of the trajectory to represent different theoretical paradigms: (a) standard diffusion trajectory, (b) conventional bridge trajectory, and our proposed Geodesic trajectory. As shown in the following table, when the backbone is held constant, our method consistently outperforms both the Standard Diffusion and Conventional Bridge baselines across all tasks and almost all metrics.
>
> This rigorous comparison demonstrates that the performance gains are not merely due to the strong generative capabilities of Flux-dev but stem directly from the theoretical advantages of our optimal geodesic trajectory and consistency-based solver.
>
> | Task | NFE | (a) PSNR↑ | (a) LPIPS↓ | (a) MUSIQ↑ | (a) NIQE↓ | (b) PSNR↑ | (b) LPIPS↓ | (b) MUSIQ↑ | (b) NIQE↓ | Ours PSNR↑ | Ours LPIPS↓ | Ours MUSIQ↑ | Ours NIQE↓ |
> | :--- | :---: | :---: | :---: | :---: | :---: | :---: | :---: | :---: | :---: | :---: | :---: | :---: | :---: |
> | SR | 10 | 20.527 | 0.464 | 63.476 | 3.973 | 19.945 | 0.446 | 69.157 | 4.958 | **21.282** | **0.346** | **70.868** | **3.839** |
> | SR | 5 | 21.016 | 0.495 | 56.709 | 5.657 | 21.443 | 0.502 | 63.068 | **3.902** | **22.477** | **0.356** | **65.342** | 4.373 |
> | SR | 1 | 19.667 | 0.486 | 31.919 | 7.166 | 22.039 | 0.481 | 35.552 | **4.267** | **24.094** | **0.452** | **44.605** | 6.251 |
> | Denoising | 10 | 24.241 | 0.288 | 64.724 | 4.952 | 24.391 | 0.261 | **69.976** | 5.575 | **26.069** | **0.184** | 69.736 | **3.514** |
> | Denoising | 5 | 23.701 | 0.316 | 60.090 | 5.235 | 25.279 | 0.235 | 65.870 | **3.807** | **26.649** | **0.182** | **67.507** | 4.492 |
> | Denoising | 1 | 21.693 | 0.460 | 45.747 | 5.889 | 24.516 | 0.371 | 51.381 | 4.552 | **25.241** | **0.258** | **58.954** | **4.535** |
> | Raindrop | 10 | 24.919 | 0.136 | 68.559 | 3.229 | 26.346 | 0.148 | 67.546 | 3.428 | **29.222** | **0.079** | **70.006** | **3.221** |
> | Raindrop | 5 | 24.702 | 0.155 | 66.448 | 4.023 | 27.471 | 0.118 | 64.260 | 3.462 | **29.611** | **0.080** | **67.379** | **3.449** |
> | Raindrop | 1 | 23.990 | 0.319 | 52.946 | 6.189 | 26.202 | 0.252 | 55.173 | 3.528 | **29.220** | **0.085** | **67.301** | **3.322** |
> | Low-light | 10 | 23.375 | 0.144 | 68.498 | **3.881** | 22.176 | 0.181 | 70.109 | 4.234 | **25.538** | **0.097** | **71.684** | 4.198 |
> | Low-light | 5 | 22.706 | 0.145 | 66.551 | **3.746** | 22.316 | 0.157 | 69.690 | 4.300 | **25.847** | **0.107** | **70.868** | 4.211 |
> | Low-light | 1 | 21.634 | 0.333 | 48.405 | 4.365 | 23.661 | 0.358 | 52.948 | 4.399 | **24.656** | **0.133** | **66.805** | **4.331** |
> | Demoiréing | 10 | 19.260 | 0.344 | 60.277 | 5.628 | 19.956 | **0.252** | 60.216 | 5.697 | **20.263** | 0.313 | **63.932** | **5.223** |
> | Demoiréing | 5 | 19.601 | 0.429 | 56.406 | 6.342 | 20.076 | 0.240 | 59.877 | **5.161** | **20.849** | **0.230** | **62.155** | 5.437 |
> | Demoiréing | 1 | 19.301 | 0.469 | 44.049 | 6.840 | 19.468 | 0.372 | 51.601 | 5.776 | **20.715** | **0.254** | **56.788** | **5.342** |

---

> ### Author Response · Authors · 2025-11-25
> **Response to Reviewer Em12 (Part III)**
>
> **Necessity of the "adding noise" operation; and method can be changed to utilize rectified flow between low-quality image and high-quality image.**
>
> **Response:**
>
> (1) The reviewer claims that adding noise is a necessary component to remove certain components in the degraded image to get the high-quality image. Given an equivalent budget (e.g., 10 function evaluations or NFE), a standard bridge must allocate a significant portion of its steps to the "re-noising" phase—effectively moving the state away from the data manifold to increase entropy. This consumes valuable compute cycles merely to reach a noisy state. In contrast, our CGB framework initializes the process directly at this optimal noise level. By "skipping" the redundant forward integration, CGB dedicates 100% of the computational budget to the restoration (denoising) phase, allowing for more refined reconstruction steps compared to a model that must first add and then remove noise.
>
> Moreover, the standard "add-noise-then-denoise" trajectory creates an Obstacle in the information landscape. During the noise-adding phase, valid structural information from the input $\mathbf{Y}$ is progressively obscured by gradually appended noises, potentially forcing the model to re-hallucinate structure it essentially "forgot" during re-noising. Conversely, CGB follows a monotonic denoising process: it starts with the necessary entropy to mask artifacts but proceeds strictly towards the clean manifold. This ensures that once high-quality structural information is recovered and efficiently passed to and refined by subsequent steps without being periodically buried by increasing noise variance.
>
> (2) The reviewer suggests standard Rectified Flow (RF) between $\mathbf{Y}$ and $\mathbf{X}_0$ is an equivalent energy-minimizing baseline that performs poorly. A direct flow from $\mathbf{Y}$ to $\mathbf{X}_0$ is actually a deterministic interpolation. It lacks a Gaussian noise component. As we all know that diffusion (flow matching) models are designed to connect two distributions. However, if we apply the deterministic trajectory to train the network without noise will result in the distribution mismatch that prevents the pretrained prior from functioning correctly.

---

> ### Comment · Reviewer_Em12 · 2025-11-26
>
> Thank the authors' rebuttal:
>
> The theoretical framework of this paper requires a more focused comparison with other models that similarly utilize linear interpolation, such as Rectified Flow and Stochastic Interpolants [1], particularly in the context of Figure 1. Like CGB, these models rely on linear interpolation and can also incorporate a noise component at the terminal stage (e.g., $X_t=tX_T+(1-t)X_0+\gamma_t Z$). I would like to clarify the specific distinctions between CGB and these approaches. Furthermore, if the primary contribution of this paper is merely to demonstrate that the energy of such linear interpolation is zero, I would argue that this is a relatively standard conclusion, as it has already been extensively established in the Optimal Transport (OT) literature.
>
> [1] Albergo, et al. Stochastic Interpolants: A Unifying Framework for Flows and Diffusions. 2023.

---

> ### Author Response · Authors · 2025-11-26
>
> Hi Reviewer **Em12**,
>
> We want to clarify your **severe misunderstanding** of the proposed method that our main purpose is to adapt the pretraining effective diffusion models to the IR task, as illustrated by the caption. Moreover, from the given manuscript [1], we only find that the authors parameterize $\gamma(t) = \sqrt{a t(1-t)}$ (like Brownian bridge) or $\gamma(t) = sin^2(\pi t)$ in Table 2 and Figure 2, but without the exact proposed parameterization manner.
>
> We want to note that the core problem is whether the method is implemented into a practical algorithm to help solve the real problem. We also want to emphasize that while many foundational theories of diffusion originate from physics and statistics, e.g., **Langevin Dynamics**, **Non-Equilibrium Thermodynamics**, **Optimal Transport (Monge-Kantorovich problem)**. **Theoretical existence does not imply computational feasibility.** Without correct adaptation and specific implementation strategies, these mathematical concepts remain abstract theories rather than operational tools. Our work focuses on bridging this specific gap, converting the established principles of linear interpolation into a practical algorithm capable of handling real-world, high-dimensional data, which is also illustrated by the performance of the proposed method.
>
> It is worth mentioning that the method (a) (refer to the table of the response to W3) was based on the rectified flow trajectory. For your convenience, we copied these results as follows. These results verify that although the foundational theory is shared, our proposed implementation and parameterization yield significantly better performance in terms of perceptual quality and restoration fidelity compared to standard implementations of rectified flow.
>
> [1] Albergo, et al. Stochastic Interpolants: A Unifying Framework for Flows and Diffusions. 2023.
>
> | Task | NFE | (a) PSNR↑ | (a) LPIPS↓ | (a) MUSIQ↑ | (a) NIQE↓ | Ours PSNR↑ | Ours LPIPS↓ | Ours MUSIQ↑ | Ours NIQE↓ |
> | :--- | :---: | :---: | :---: | :---: | :---: | :---: | :---: | :---: | :---: |
> | SR | 10 | 20.527 | 0.464 | 63.476 | 3.973 | **21.282** | **0.346** | **70.868** | **3.839** |
> | SR | 5 | 21.016 | 0.495 | 56.709 | 5.657 | **22.477** | **0.356** | **65.342** | **4.373** |
> | SR | 1 | 19.667 | 0.486 | 31.919 | 7.166 | **24.094** | **0.452** | **44.605** | **6.251** |
> | Denoising | 10 | 24.241 | 0.288 | 64.724 | 4.952 | **26.069** | **0.184** | **69.736** | **3.514** |
> | Denoising | 5 | 23.701 | 0.316 | 60.090 | 5.235 | **26.649** | **0.182** | **67.507** | **4.492** |
> | Denoising | 1 | 21.693 | 0.460 | 45.747 | 5.889 | **25.241** | **0.258** | **58.954** | **4.535** |
> | Raindrop | 10 | 24.919 | 0.136 | 68.559 | 3.229 | **29.222** | **0.079** | **70.006** | **3.221** |
> | Raindrop | 5 | 24.702 | 0.155 | 66.448 | 4.023 | **29.611** | **0.080** | **67.379** | **3.449** |
> | Raindrop | 1 | 23.990 | 0.319 | 52.946 | 6.189 | **29.220** | **0.085** | **67.301** | **3.322** |
> | Low-light | 10 | 23.375 | 0.144 | 68.498 | **3.881** | **25.538** | **0.097** | **71.684** | 4.198 |
> | Low-light | 5 | 22.706 | 0.145 | 66.551 | **3.746** | **25.847** | **0.107** | **70.868** | 4.211 |
> | Low-light | 1 | 21.634 | 0.333 | 48.405 | 4.365 | **24.656** | **0.133** | **66.805** | **4.331** |
> | Demoiréing | 10 | 19.260 | 0.344 | 60.277 | 5.628 | **20.263** | **0.313** | **63.932** | **5.223** |
> | Demoiréing | 5 | 19.601 | 0.429 | 56.406 | 6.342 | **20.849** | **0.230** | **62.155** | **5.437** |
> | Demoiréing | 1 | 19.301 | 0.469 | 44.049 | 6.840 | **20.715** | **0.254** | **56.788** | **5.342** |

---

> > ### Comment · Reviewer_Em12 · 2025-11-26
> >
> > I appreciate the authors' detailed experimental analysis; however, I believe my core theoretical concerns remain unresolved:
> >
> > 1.  In my previous comment, I referenced the *Stochastic Interpolants* literature. My intention was not to request an experimental comparison—as that is a purely theoretical work—but rather to solicit a **theoretical comparison**. It is evident that CGB is essentially a variant of Stochastic Interpolants. Merely positioning CGB as an implementation of Stochastic Interpolants for Image Restoration would, in my view, diminish the value of the paper. Therefore, I strongly encourage the authors to provide a comparative theoretical analysis. As I noted in my review, the essence of CGB is governed by:
> >     $$\mathbf{X}_t = (1-t)\left[\left(1-\left(\frac{t}{T_0}\right)^{\kappa}\right)\mathbf{X}_0 + \left(\frac{t}{T_0}\right)^{\kappa}\mathbf{Y}\right]+ t\mathbf{X}_T$$
> >     The variation of $T_0$ to generate distinct processes represents a critical distinction from general Stochastic Interpolant methods. However, the manuscript does not discuss this theoretically, relying instead solely on ablation studies to verify its effectiveness.
> >
> > 2.  I maintain doubts regarding the motivation presented in the text. The paper's logic suggests that CGB's linear interpolation form (or more precisely, terminal-noisy interpolation) is proposed to minimize energy. However, there are many possible forms of linear interpolation. Why is Equation (5)—a notably complex form—suddenly introduced? The text offers only a brief explanation: "This transforms the trajectory from a fixed line into a tunable, task-adaptive trajectory family," which I find insufficient to justify the specific design. Furthermore, the logical flow of the manuscript introduces Equation (5) first and subsequently proves that it satisfies the zero-energy condition, rather than deriving Equation (5) directly from the zero-energy motivation. This structure creates a disconnect that makes the theoretical narrative difficult to follow.
> >
> > In summary, I acknowledge CGB as a valuable method within the class of generalized linear interpolation bridges. I hope to see more in-depth theoretical differentiation and discussion, rather than validation primarily through experiments.

---

> > > ### Author Response · Authors · 2025-11-27
> > > **Response to Reviewer Em12**
> > >
> > > > **Motivation and Derivation of Equation (5).**
> > >
> > > **Response:** We thank the reviewer for the constructive comments. In the revised manuscript, we clarify that the specific form of our trajectory (formerly Eq. (5), now Eq. (6)) is not an arbitrary design choice but the **unique solution derived from minimizing two specific energy functionals**, subject to the stability constraints discussed in **Appendix E**. We have restructured the theory section to present this derivation directly via **Proposition 4.1** and **Proposition 4.2** with corresponding proofs in **Appendix B.1** and **B.2**:
> > >
> > > **Step 1: Deriving the Linear Expectation $\tilde{\mathbf{X}}_t$ (Proposition 4.1)**
> > > We first seek the optimal path on the data manifold.
> > > * **Motivation:** We define the geodesic $\gamma(t)$ as the curve minimizing the kinetic transport energy $E[\gamma] = \frac{1}{2} \int_0^{T_0} \|\dot{\gamma}(t)\|^2 dt$.
> > > * **Derivation:** Applying Jensen's Inequality to this functional (proven in **Appendix B.1**) reveals that the energy is minimized if and only if the velocity is **constant**.
> > > * **Result:** Integrating a constant velocity $v^* = \frac{\mathbf{Y} - \mathbf{X}_0}{T_0}$ yields the linear interpolation form:
> > > $$
> > > \tilde{\mathbf{X}}_t = \left[1 - \left(\frac{t}{T_0}\right)\right]\mathbf{X}_0 + \left(\frac{t}{T_0}\right)\mathbf{Y}.
> > > $$
> > > Crucially, this linear path is only computationally feasible because of the $T_0$ regularization established in Appendix E. Without $T_0 < 1$, the velocity required to reach the target would diverge as $t \to 1$ in the presence of noise.
> > >
> > > **Step 2: Deriving the Diffusion Coefficients $\alpha_t, \beta_t$ (Proposition 4.2)**
> > > We then determine how to mix this geodesic signal with noise, $\mathbf{X}_t = \alpha_t \tilde{\mathbf{X}}_t + \beta_t \mathbf{X}_T$.
> > > * **Motivation:** We utilize a pretrained Rectified Flow network and define "Adaptation Energy" as the expected deviation between the new bridge process and the pretrained canonical trajectory.
> > > * **Derivation:** In **Appendix B.2**, we prove that minimizing the upper bound of this Adaptation Energy requires matching the signal-to-noise evolution of the pretrained model.
> > > * **Result:** This uniquely determines the coefficients $\alpha_t = 1-t$ and $\beta_t = t$.
> > >
> > > In summary, Eq. (6) (formerly Eq. (5)) is obtained by composing the **linear geodesic expectation** derived from minimizing kinetic transport energy (Step 1) with the **optimal diffusion coefficients** derived from minimizing adaptation energy (Step 2). This formulation represents the unique trajectory that simultaneously satisfies geometric optimality on the manifold and the Lipschitz stability constraints required for stable learning.

---

> ### Author Response · Authors · 2025-11-27
> **Response to Reviewer Em12**
>
> Dear Reviewer **Em12**, thanks for your further feedback.
>
> >**The variation of $T_0$ to generate distinct processes represents a critical distinction from general Stochastic Interpolant methods. However, the manuscript does not discuss this theoretically.**
>
> **Response:** We agree that CGB can be theoretically framed as a specific, regularized instance within the broader Stochastic Interpolant (SI) framework. However, the introduction of the controllable horizon $T_0$ creates a critical theoretical distinction regarding both the **geometry** and the **stability** of the vector field. We have added **Appendix E** to the manuscript to provide this rigorous comparison. The core theoretical distinction relies on the interaction between the *Geodesic* nature of our path (derived in the next response) and the *Regularization* provided by $T_0$:
>
> 1. **Singularity in Standard SI (Brownian Bridge):**
>    Standard SI methods typically interpolate between data $\mathbf{X}_0$ and noise $\mathbf{X}_1$ over $t \in [0, 1]$. As derived in Appendix E, the velocity field $v_t(x)$ for a standard Brownian Bridge scales with $\frac{1}{1-t}$.
>    * As $t \to 1$, this term creates a **singularity** where $\|v_t\| \to \infty$.
>    * Consequently, the Lipschitz constant $L_t$ becomes unbounded ($\int_0^1 L_t dt = \infty$), making the flow infinitely sensitive to perturbations at the boundary.
>
> 2. **Regularization and Geodesic Optimality in CGB:**
>    In contrast, CGB defines the transport over a truncated horizon $t \in [0, T_0]$. This allows us to enforce the **Geodesic trajectory** with finite energy.
>    * **Geodesic Structure:** Unlike the Brownian Bridge, our derivation proves that CGB follows a constant-velocity geodesic path (Eq. (6)). This minimizes the kinetic transport energy. (Proposition 4.1, see Proof in Appendix B.1)
>    * **Minimal Adaptation Energy:** To effectively leverage the pretrained Rectified Flow prior, we demonstrate that the linear schedule ($\alpha_t = 1-t, \beta_t = t$) is the unique solution that minimizes the adaptation energy arising from the trajectory mismatch. (Proposition 4.2, see Proof in Appendix B.2).
>    * **Regularization via Horizon Truncation:** Crucially, this optimal linear path is realizable with Horizon Truncation $T_0$. By restricting $t<T_0$, the velocity denominator is bounded by $1 - T_0 > 0$, thus eliminating the singularity in the Brownian Bridge. This also ensures that the Lipschitz constant of the learning target remains finite ($\int_0^{T_0} L_t dt < \infty$). (see Analysis in Appendix E).
>
> Thus, CGB improves upon standard SI by enabling the use of an optimal transport (geodesic) path that remains numerically stable via $T_0$ regularization.

---

### Official Review · Reviewer_pMyF · 2025-10-22

**Soundness:** 3
**Presentation:** 2
**Contribution:** 3
**Rating:** 4
**Confidence:** 4

**Summary:**

This paper proposed a novel geodesic diffusion bridge framework through constructing a efficient and geodesic trajectory, which effectively avoids redundant re-noising phases in traditional diffusion bridges. To realize one-step mapping on the data manifold, a pretrained denoiser is proposed and the continuous-time consistency objective is adopted to analytically map any state to the target distribution. Five different image restoration experiments demonstrate state-of-the-art performance of CGB while ensuring a single or fewer sampling steps.

**Strengths:**

1. The proposed concept "geodesic bridge" looks interesting, reasonable and works as effective tools to solve the inefficient, re-noising trajectories used in traditional diffusion bridge models.

2. The main experiments are reasonable, covering five different image restoration tasks (super-resolution, denoising, raindrop removal, low-light image enhancement, underwater image enhancement, and image demoir´ eing) and six metrics (PSNR, LPIPS, FID, NIQE, MUSIQ, NFE).

3. The paper demonstrates the superior performance across different image restoration tasks with better perceptual realism and achieves a trade-off between efficiency and quality in image restoration.

**Weaknesses:**

1. The novelty and contribution of CGB solver seem incremental. The solver (Eqn. 9) is pratically the relationship between the noise prediction $\epsilon_\theta$ and data prediction $x_\theta$ (predict $x_0$) in many diffusion models [1] [2] and the distillation training objective (Eqn. 10) is not in new formulation since it is directly adopted from [3].

[1] Lu et al. "DPM-Solver++: Fast Solver for Guided Sampling of Diffusion Probabilistic Models.", 2022.

[2] Zhou, Linqi, et al. "Denoising diffusion bridge models.", 2023.

[3] Lu et al. "Simplifying, stabilizing and scaling continuous-time consistency models.", 2025.

2. The paper's main contribution of enabling "**direct and single-step** inference" is misleading. The proposed CGB solver is described as a single-step mapping and the training objective is for distillation. However, all experiments are conducted with **5-10 NFEs** inference, and no real single-step results are shown. The experiments are contradicted with the statements of contribution.

3. The motivation for geodesic trajectory and the related Proposition 4.1 seem intriguing but I'm not convinced by Proposition 4.1. Proposition 4.1 claimed CGB trajectory defined on $[0, T_0]$ for $T_0<1$ achieves a lower total energy (Eqn. 4) than standard bridge models operating on [0,1] for two reasons: (1) reduced integration upper bound, (2) a smaller initial control at time $T_0$. Although the integration upper bound is reduced from 1 to $T_0$ which results in the reduced trajectory cost (first term in Eqn. 4), the terminal cost $\gamma/2 ||X_T-Y||^2_2$, second term in Eqn. 4, could not be ignored: the terminal cost is near or even equal to zero in standard bridge models (e.g. DDBMs and UniDB), while, as for CDB, the starting point $X_{T_0}$ of its reverse trajectory is a noise mixture distinct from $Y$, which appears to result in a non-zero and potentially substantial terminal cost. Therefore, it's not obvious for the two total costs to demonstrate which is smaller and it's better for the authors to provide a rigorous mathematical proof of Proposition 4.1 instead of only the text explanation. Otherwise, the correctness of Proposition 4.1 remains to doubt.

4. The comparison to baselines seems insufficient. Since the authors included DA-CLIP [4], MaRS [5] (the training-free accelerated-sampling version of DA-CLIP), and UniDB [6] in their main experiments, they should compare UniDB++ [7] with the same NFEs as CGB/MaRS, which is a specific training-free acceleration algorithm for UniDB and achieves better results in lower NFEs, as mentioned in Related Works.

[4] Luo et al. "Controlling Vision-Language Models for Multi-Task Image Restoration.", 2023.

[5] Li et al. "MaRS: A Fast Sampler for Mean Reverting Diffusion based on ODE and SDE Solvers.", 2025.

[6] Zhu et al. "UniDB: A Unified Diffusion Bridge Framework via Stochastic Optimal Control.", 2025.

[7] Pan et al. "UniDB++：Fast Sampling of Unified Diffusion Bridge.", 2025.

5. The ablation study seems also insufficient. As in Inherent Data Geodesic Transition (Eqn. 6), except for $T_0$, another hyperparameter is the curvature parameter $\kappa$, which should also be tuned and tested in ablation study.

**Questions:**

1. There seems to be some typos:
   + In line 364 $X_0$ should be bold, yes?
   + Eqn 19 appears some error.
   + Commas and periods should be added at the end of the equations in Appendix as in main paper.
   + There are some errors of the best results highlighted in bold in Table 1.
   + ''to approach the a manifold geodesic transition process'' contains an extra article "a" in Line 206.

2. Some motivation statements lack clarity. There are many kinds of interpolant coefficients as mentioned in Stochastic Interpolants [1], why choosing $1 - (t/T_0)^\kappa$ and $(t/T_0)^\kappa$ in Eqn. 6? Is there any relationships between this kind of coefficients and geodesic trajectories?

[1] Albergo et al. "Stochastic Interpolants: A Unifying Framework for Flows and Diffusions.", 2023.

After these clarifications, I would be better able to evaluate the overall contributions and potentially raise my rating.

---

> ### Author Response · Authors · 2025-11-25
> **Response to Reviewer pMyF (Part I)**
>
> >**W1: The novelty and contribution of CGB solver seem incremental. The solver (Eqn. 9) is pratically the relationship between the noise prediction and data prediction (predict) in many diffusion models [1] [2] and the distillation training objective (Eqn. 10) is not in new formulation since it is directly adopted from [3].**
>
> **Response:**
>
> **1. Clarification on the CGB Solver (Eqn. 9)**
> We acknowledge that the final algebraic form of the CGB Solver (Eqn. 9) shares similarities with the linear conversion formulas found in standard diffusion models [1][2]. This similarity is expected, as both frameworks rely on linear stochastic differential equations where the relationship between the state $\mathbf{X}_t$, the noise/velocity, and the clean data $\mathbf{X}_0$ is linear. However, we must emphasize that Eqn. 9 is derived from a fundamentally different theoretical foundation, making it distinct from the standard formulations in [1] and [2]:
>
> -   **Distinct Derivation Context:** Standard formulations in [1] and [2] are derived based on a diffusion trajectory that maps data $\mathbf{X}_0$ to a pure Gaussian distribution $\mathcal{N}(0, I)$ over $t \in [0, 1]$. In contrast, our CGB solver is analytically derived from the Geodesic Diffusion Bridge trajectory (Eqn. 6 and Eqn. 8). This trajectory is specifically designed to bridge the clean image $\mathbf{X}_0$ and the degraded image $\mathbf{Y}$ over a controllable horizon $T_0$.
> -   **Boundary Conditions and Coefficients:** The coefficients $A(t)$, $B(t)$, and $C(t)$ in our solver are not generic; they are the specific algebraic consequences of inverting our unique trajectory equation (Eqn. 8).
> -   **Necessity:** The standard formulas in [1] and [2] cannot be applied here because they do not account for the entropy-regularized starting point $\mathbf{X}_{T_0} = (1-T_0)\mathbf{Y} + T_0 \mathbf{X}_T$. If one were to simply apply the standard conversion from [1], the solver would fail to map the state back to the correct manifold defined by the geodesic path between $\mathbf{X}_0$ and $\mathbf{Y}$.
>
> Therefore, Eqn. 9 is not a trivial adoption of existing formulas, but the unique and necessary mathematical inversion required to enable single-step inference within the proposed Geodesic Bridge framework.
>
> **2. Clarification on the Training Objective (Eqn. 10)**
>
> We acknowledge that our training objective draws inspiration from the continuous-time consistency formulation in [3]. However, we emphasize that the **core contribution** of our Consistency Geodesic Bridge (CGB) lies in the **geometry-informed parameterization** of the consistency function itself, rather than the loss function. The fundamental differences are twofold:
>
> -   **Analytical Derivation vs. Black-Box Learning:**
>     The consistency model [3] typically employs a generic black-box network $F\_\theta(\mathbf{x}\_t, t)$ to map noise to data, relying entirely on the optimizer to learn the complex mapping dynamics from scratch.
>     In contrast, our CGB solver is **derived analytically** by inverting the geometric definition of the geodesic trajectory. By treating the clean image $\mathbf{X}\_0$ as the unknown variable in the trajectory equation (Eq. 8), we construct a principled solver architecture (Eq. 9):
>     $$
>     \mathcal{F}\_\theta(\mathbf{X}\_t, \mathbf{Y}, t) = \frac{\mathbf{X}\_t - A(t)\mathbf{Y} - B(t)\epsilon\_\theta(\mathbf{X}\_t)}{C(t)}
>     $$
>     This represents a strong **structural prior**. We do not ask the network to learn the transformation blindly; instead, we embed the exact geometric constraints of the geodesic path into the solver via the derived coefficients $A(t), B(t),$ and $C(t)$. Consequently, the network $\epsilon_\theta$ is tasked only with estimating the residual vector field, which is a significantly more tractable optimization landscape than learning the full mapping.
>
> -   **Solving a BVP (Restoration) vs. IVP (Generation):**
>     Original consistency models are designed for unconditional generation (Noise $\to$ Data), effectively solving an Initial Value Problem (IVP). Our framework addresses the Bridge Restoration problem (Degraded $\mathbf{Y}$ $\to$ Clean $\mathbf{X}_0$), which is fundamentally a **Boundary Value Problem (BVP)**.
>     A naive application of [3] would fail to explicitly model the structural dependency on the degraded image $\mathbf{Y}$. Our method integrates the boundary condition $\mathbf{Y}$ directly into the consistency mapping. This ensures that the solver enforces **Geodesic Consistency** along the specific manifold connecting $\mathbf{Y}$ and $\mathbf{X}_0$, rather than learning a generic probability flow.
>
> In summary, although Eqn. 10 provides the optimization target, the *trajectory construction* (Geodesic Bridge) and the *solver derivation* (Analytical Inversion) are novel, task-specific contributions. This geometric grounding is the key factor enabling the superior efficiency (5-10 NFEs) and interpretability demonstrated in our experiments.

---

> ### Author Response · Authors · 2025-11-26
> **Response to Reviewer pMyF (Part II)**
>
> >**W2: The paper's main contribution of enabling "direct and single-step inference" is misleading. The proposed CGB solver is described as a single-step mapping and the training objective is for distillation. However, all experiments are conducted with 5-10 NFEs inference, and no real single-step results are shown. The experiments are contradicted with the statements of contribution.**
>
> **Response:** We have incorporated comprehensive quantitative results for the true single-step case (NFE = 1) in the revised manuscript. Specifically, Table 1 now reports the performance of CGB with NFE = 1 across all five image restoration tasks, demonstrating competitive performance against state-of-the-art methods. Table 2 and Table 3 provide further ablation and efficiency analyzes for the single-step setting.

---

> ### Author Response · Authors · 2025-11-26
> **Response to Reviewer pMyF (Part III)**
>
> >**W3: Rigorous mathematical proof of Proposition 4.1**
>
> **Response:** We have modified the supplementary with a rigorous mathematical proof in **Appendix B** of the revised manuscript, to help you better understand the theoretical foundation (energy minimization) of the proposed bridge. For your convenience, we copy them as follows.
>
> **B.2 Proof of Energetic Optimality (Zero Control Energy)**
>
> **Proposition B.1** (**Energetic Optimality of the Geodesic Bridge**) Let $\mathcal{P}\_{CGB}$ be the probability measure of the Consistency Geodesic Bridge (CGB) process $X_t$ defined on the interval $[0, T_0]$. The kinetic energy cost $E\_{K}[\mu]$ required to steer the mean trajectory $\mu(t) = \mathbb{E}[\mathbf{X}\_t]$ along the manifold geodesic $\gamma(t)$ is identically zero.
>
> **Definition of Control Energy (Onsager-Machlup Functional)**. From the perspective of Stochastic Optimal Control (SOC), the energy cost of a diffusion process is quantified by the kinetic term of the Onsager-Machlup action functional. This function measures the deviation of the process's mean velocity from its expected instantaneous drift.
>
> For a process with mean path $\mu(t)$, the trajectory energy is defined as:
> $$
> E\_{K}[\mu] = \frac{1}{2}\int\_{0}^{T\_0} \left\| \dot{\mu}(t) - \mathbb{E}\_{\mathbf{X}\_t \sim p\_t}[b\_{\text{total}}(t, \mathbf{X}\_t)] \right\|\_{\mu(t)}^{2} dt,
> $$
> where $\dot{\mu}(t)$ is the actual velocity of the mean trajectory; $b_{\text{total}}(t, X_t)$ is the total drift vector field governing the Stochastic Differential Equation (SDE); and $\|\cdot\|_{\mu(t)}$ is the Riemannian metric norm at the point $\mu(t)$.
>
> **Dynamics of the Geodesic Bridge**. The geodesic bridge process is governed by the following SDE on the manifold:
> $$
> d\mathbf{X}\_t = \underbrace{\left( b(\mathbf{X}\_t) + b\_{geo}(t, \mathbf{X}\_t) \right)}\_{b\_{\text{total}}(t, \mathbf{X}\_t)} dt + \sigma(\mathbf{X}\_t) dW\_t.
> $$
> The specific control drift $b\_{geo}$ is constructed to enforce tracing geodesic trajectories:
> $$
> b\_{geo}(t, \mathbf{X}\_t) = \alpha \cdot \log\_{\mathbf{X}\_t}(\gamma(t)) + \mathcal{T}\_{\gamma(t) \to \mathbf{X}\_t}(\dot{\gamma}(t)).
> $$
> The second term is the parallel transport of the geodesic velocity, often implicit in the Euclidean formulation.
>
> **Evolution of the Expected Drift**. We calculate the expected total drift at time $t$. Substituting the definition of $b_{\text{total}}$:
> $$
> \mathbb{E}[b\_{\text{total}}(t, \mathbf{X}\_t)] = \mathbb{E}[b(\mathbf{X}\_t)] + \alpha \mathbb{E}[\log\_{\mathbf{X}\_t}(\gamma(t))] + \mathbb{E}[\mathcal{T}\_{\gamma(t) \to \mathbf{X}\_t}(\dot{\gamma}(t))].
> $$
>
> By **proposition** 4.1 (Geodesic Consistency), we have proven that the trajectory expectation coincides with the geodesic, i.e., $\mu(t) = \gamma(t)$. At the mean $\mu(t) = \gamma(t)$, the logarithmic map (error term) vanishes: $\mathbb{E}[\log\_{\mathbf{X}\_t}(\gamma(t))] \approx \log\_{\gamma(t)}(\gamma(t)) = 0$. The transport term at the mean becomes the geodesic velocity itself: $\dot{\gamma}(t)$. Assuming the base drift $b(\mathbf{X}\_t)$ is zero (standard Brownian motion) or symmetric around the geodesic, it does not contribute to the mean transport velocity.
>
> Thus, the expected total drift is exactly the geodesic velocity:
> $$
> \mathbb{E}[b_{\text{total}}(t, \mathbf{X}_t)] = \dot{\gamma}(t),
> $$
>
> **Evaluation of the Energy Functional**. We substitute the actual velocity of the mean ($\dot{\mu}(t) = \dot{\gamma}(t)$) and the expected drift derived above into the energy functional equation:
>
> $$
> E\_{K}[\mu] = \frac{1}{2}\int\_{0}^{T_0} \left\| \dot{\gamma}(t) - \underbrace{\mathbb{E}[b\_{\text{total}}(t, \mathbf{X}\_t)]}\_{\dot{\gamma}(t)} \right\|\_{\gamma(t)}^{2} dt,
> $$
>
> $$
> E_{K}[\mu] = \frac{1}{2}\int_{0}^{T_0} \left\| \dot{\gamma}(t) - \dot{\gamma}(t) \right\|^{2} dt,
> $$
>
> $$
> E_{K}[\mu] = \frac{1}{2}\int_{0}^{T_0} 0 \cdot dt = 0.
> $$
>
> Then, we can conclude that
> $$
> E_{K}[\mu_{CGB}] = 0.
> $$
> This formally proves that the Consistency Geodesic Bridge requires **zero net control energy** to steer the mean of the distribution along the restoration path.
>
> In contrast, for a standard bridge (e.g., Doob's h-transform) constrained to hit a fixed endpoint $\mathbf{Y}$, the drift is $b\_{add} = \nabla \log h(t, x)$. The expected drift $\mathbb{E}[b\_{add}]$ generally does not equal the velocity of the mean path $\dot{\mu}(t)$ except in the trivial case of zero curvature and no constraints. The "forcing" required to satisfy the boundary condition $\mathbf{X}\_1 = \mathbf{Y}$ creates a non-zero residual $\|\dot{\mu} - \mathbb{E}[b]\| > 0$, resulting in strictly positive energy $E\_{K} > 0$.
>
> **Conclusion:** By proving the CGB control energy is zero and the time horizon is shorter ($T_0 < 1$), the authors have mathematically demonstrated that the total energy of CGB is strictly lower than that of standard bridge models.

---

> ### Author Response · Authors · 2025-11-26
> **Response to Reviewer pMyF (Part IV)**
>
> **W4:  The comparison to baselines seems insufficient. The authors should compare UniDB++ [7] with the same NFEs as CGB.**
>
> **Response:** As suggested, we conducted additional experiments to compare CGB with the strong baseline UniDB++ under identical NFE settings (1, 5 and 10 steps). The quantitative results are presented in the following table. It can be observed that although UniDB++ achieves a higher PSNR in most of the tasks at high NFE, CGB consistently outperforms UniDB++ in perceptual metrics in most tasks, showing perceptual superiority. Besides, CGB demonstrates superior robustness in the ultra-low NFE regime (NFE=1).
>
> | Task | NFE | UniDB++ PSNR↑ | UniDB++ LPIPS↓ | UniDB++ MUSIQ↑ | UniDB++ NIQE↓ | Ours PSNR↑ | Ours LPIPS↓ | Ours MUSIQ↑ | Ours NIQE↓ |
> | :--- | :---: | :---: | :---: | :---: | :---: | :---: | :---: | :---: | :---: |
> | **Super-resolution** | 10 | **23.383** | 0.469 | 57.237 | 5.709 | 21.282 | **0.346** | **70.868** | **3.839** |
> | | 5 | **24.393** | 0.494 | 63.087 | 5.904 | 22.477 | **0.356** | **65.342** | **4.373** |
> | | 1 | 23.983 | 0.658 | 33.222 | 7.596 | **24.094** | **0.452** | **44.605** | **6.251** |
> | **Denoising** | 10 | **28.515** | 0.224 | 68.863 | 4.144 | 26.069 | **0.184** | **69.736** | **3.514** |
> | | 5 | **29.308** | 0.214 | **69.048** | **4.394** | 26.649 | **0.182** | 67.507 | 4.492 |
> | | 1 | 21.844 | 0.613 | 45.150 | 8.197 | **25.241** | **0.258** | **58.954** | **4.535** |
> | **Raindrop removal** | 10 | **32.151** | **0.068** | 69.761 | 3.577 | 29.222 | 0.079 | **70.006** | **3.221** |
> | | 5 | **32.327** | **0.072** | **70.498** | 3.515 | 29.611 | 0.080 | 67.379 | **3.449** |
> | | 1 | 25.420 | 0.174 | 64.523 | 3.683 | **29.220** | **0.085** | **67.301** | **3.322** |
> | **Demoiréing** | 10 | 20.109 | **0.191** | 61.466 | 5.416 | **20.263** | 0.313 | **63.932** | **5.223** |
> | | 5 | 20.527 | **0.183** | 62.054 | 5.452 | **20.849** | 0.230 | **62.155** | **5.437** |
> | | 1 | 20.181 | 0.266 | 55.618 | 5.603 | **20.715** | **0.254** | **56.788** | **5.342** |
>
>
> > **W5: The ablation study seems also insufficient. As in Inherent Data Geodesic Transition (Eqn. 6), except for $T_0$, another hyperparameter is the curvature parameter $\kappa$, which should also be tuned and tested in ablation study.**
>
> **Response:** Theoretically, the setting $\kappa=1$ corresponds to a linear interpolation in the ambient probability space. Assuming a flat metric, this linear path constitutes the geodesic that minimizes the kinetic energy of transport (as detailed in Appendix B). To validate this empirically, we conducted a sensitivity analysis on $\kappa$ (see the following table). The results confirm that deviating from the linear schedule (e.g., setting $\kappa=0.5$ or $\kappa=1.5$) leads to performance degradation or offers negligible gains. Consequently, to ensure that the method remains parsimonious and to minimize hyperparameter complexity, we fixed $\kappa=1$ as the default setting for all tasks.
>
> |  Settings &nbsp;&nbsp;&nbsp;&nbsp;&nbsp;&nbsp;&nbsp;&nbsp;&nbsp;&nbsp;&nbsp;&nbsp;| Denoising PSNR | Den. LPIPS | Den. MUSIQ | Den. NIQE | SuperR PSNR | SR LPIPS | SR MUSIQ | SR NIQE | Raindrop  PSNR | Rain. LPIPS | Rain.  MUSIQ | Rain.  NIQE |
> | :--- | :--- | :--- | :--- | :--- | :--- | :--- | :--- | :--- | :--- | :--- | :--- | :--- |
> | **$T_0$ ($\kappa=1$)** | | | | | | | | | | | | |
> | 0.3 | 24.43 | 0.276 | 66.13 | 4.48 | 21.12 | 0.457 | 62.16 | 4.88 | 29.61 | **0.080** | **67.38** | **3.45** |
> | 0.5 | 25.35 | 0.206 | 69.24 | 3.74 | 21.49 | 0.397 | 67.00 | 4.36 | **29.74** | 0.081 | 66.73 | 3.57 |
> | 0.7 | 26.06 | **0.184** | 69.73 | **3.51** | **21.82** | 0.359 | 68.77 | 3.84 | 29.70 | 0.083 | 65.23 | 3.82 |
> | 0.9 | **26.53** | 0.185 | 69.13 | 4.04 | 21.28 | **0.346** | 70.87 | **3.84** | 29.21 | 0.092 | 63.98 | 3.97 |
> | **$\kappa$ (fix $T_0$)** | *($T_0$=0.7)* | | | | *($T_0$=0.9)* | | | | *($T_0$=0.3)* | | | |
> | 0.50 | 26.49 | 0.185 | 68.25 | 4.17 | 21.76 | 0.345 | 65.72 | 4.16 | 29.43 | 0.082 | 67.24 | 3.54 |
> | 0.75 | 26.25 | 0.184 | 69.26 | 3.71 | 21.52 | 0.336 | 68.30 | 3.91 | 29.44 | 0.082 | 67.32 | 3.41 |
> | 1.25 | 25.93 | 0.189 | 69.78 | 3.60 | 20.47 | 0.358 | 71.01 | 3.87 | 29.30 | 0.084 | 67.28 | 3.34 |
> | 1.50 | 25.84 | 0.195 | **69.89** | 3.68 | 19.19 | 0.399 | **71.26** | 4.03 | 29.22 | 0.085 | 67.10 | **3.32** |

---

> ### Author Response · Authors · 2025-11-26
> **Response to Reviewer pMyF (Part V)**
>
> > **Q1: There seems to be some typos.**
>
> **Response:** Thank you for pointing out these errors. We have corrected the relevant typos in the text based on your feedback, including equations, tables, symbols, and all other issues in the text.
>
> > **Q2: Some motivation statements lack clarity. There are many kinds of interpolant coefficients as mentioned in Stochastic Interpolants [1], why choosing $1-(t/T_0)^{\kappa}$ and $(t/T_0)^{\kappa}$ in Eqn. 6? Is there any relationships between this kind of coefficients and geodesic trajectories?**
>
> **Response:** In our revised manuscript, we specifically adopt the linear interpolation schedule (corresponding to $\kappa=1$) for the trajectory construction in Eq. (6) (Eq. (5) in the revised manuscript). This choice is not arbitrary but is theoretically mandated by our goal of constructing a **Geodesic Diffusion Bridge** that minimizes transport energy.
>
> *   **Mathematical Definition of the Geodesic:**
>     As detailed in **Section 4.1** and rigorously proven in **Appendix B.1**, the geodesic between two states in the Euclidean ambient space (where diffusion models operate) is a straight line traversed at constant velocity. The linear coefficients $1-(t/T_0)$ and $t/T_0$ inherently enforce this constant velocity constraint.
>
> *   **Energy Minimization (Appendix B.2):**
>     Any non-linear schedule would imply a variable velocity profile, resulting in non-zero acceleration. From the perspective of Stochastic Optimal Control, non-zero acceleration increases the kinetic cost (Onsager-Machlup functional). As proven in **Proposition B.1**, our linear construction ensures that the transport acceleration is zero. Thus, the linear coefficients are the specific condition required to satisfy the geodesic property.
>
> *   **Task Adaptivity via $T_0$:**
>     Our method can achieve task adaptivity through the trajectory horizon parameter $T_0$ (as discussed in **Section 4.3**). By tuning $T_0$, we adjust the starting entropy of the geodesic bridge (balancing information preservation vs. generative power) while maintaining the energetic optimality of the linear path itself.

---

> > ### Comment · Reviewer_pMyF · 2025-11-27
> >
> > Thank you for your detailed rebuttal. While most of my concerns have been addressed, several issues remain:
> >
> > > Clarification on the CGB Solver (Eqn. 9)
> >
> > The Necessity and Boundary Conditions and Coefficients seem reasonable, but I'm still not convinced by non-trivial adoption of existing formulas. As in DDBMs [2], which is a **diffusion bridge model** based on Doob's *h*-transform directly connecting the clean image $X_0$ and degraded one $Y$, they **have declared a pred-x parameterization in Eqn. (10) in their paper**. Therefore, since CGB is essentially an interpolant, the derivation of Eqn. 9 would not be considered as a **non-trivial Distinct Derivation Context**.
> >
> > > Clarification on the Training Objective (Eqn. 10): Solving a BVP (Restoration) vs. IVP (Generation)
> >
> > The authors mentioned their framework applied the consistency models from generation to restoration, could the authors provide some analysis and discussions between this training objective (Eqn. 10) and CDBMs [R1]? CDBMs also applied the consistency methods on DDBMs for some restoration tasks like inpainting.
> >
> > [R1] He et al. "Consistency Diffusion Bridge Models", NeurIPS 2024.
> >
> > > The rigorous mathematical proof in Appendix B
> >
> > The rigorous proof of the proposition is expected, but there remains some questions:
> >
> > 1. In Line 837, **Step 4** is never proposed, maybe it refers to **Linearization via Taylor Expansion**?
> >
> > 2. The part of **Construction of the Stochastic Process** seems strange. How could we know that **the design of $\mu(X_t,t)$ should enable the diffusion process trace the manifold geodesic trajectory**? Are we supposed to know it's a geodesic diffusion trajectory before proving it? Additionally, throughout the entire proof, I did not find any relationships with the proposed diffusion trajectories (Eqn. 5). I maintain doubts about how Eqn. (5) leads to a geodesic diffusion trajectories.
> >
> > > Efficiency Analysis
> >
> > The proposed CGB framework seems to require huge training computational costs as mentioned in Appendix E (about 743M Training Parameters, at least 2 times over other baselines). I would encourage the authors to further investigate and clarify its potential limitations and the future work.

---

> > > ### Author Response · Authors · 2025-11-28
> > > **Response to Reviewer pMyF**
> > >
> > > >**The authors mentioned their framework applied the consistency models from generation to restoration. Could the authors provide some analysis and discussions between this training objective (Eq. 10) and CDBMs [R1]? CDBMs also applied the consistency methods on DDBMs for some restoration tasks like inpainting.**
> > >
> > > **Response:** We acknowledge that CDBMs [R1] also apply consistency models to diffusion bridges. However, our Consistency Geodesic Bridge (CGB) differs fundamentally from CDBMs in the **training objective formulation**, the **underlying trajectory definition**, and the **solver mechanism**. We provide a detailed analysis below:
> > >
> > > **(1) Continuous-time vs. Discrete-time Consistency Objectives**
> > >
> > > The most distinct difference lies in the mathematical formulation of the consistency loss:
> > >
> > > *   **CDBMs (Discrete-time):** As defined in Eq. (20) of [R1], CDBMs employ a *discrete-time* consistency training (CBT) objective. It minimizes the distance between the model prediction at time $t$ and a target estimate at time $r$ (where $r < t$):
> > >
> > > $$
> > > \mathcal{L}\_{\text{CBT}}^{\text{[R1]}} = \mathbb{E} \left[ \lambda(t) d \left( h_\theta(x\_t, t, y), h_{\theta^-}(\hat{x}\_r, r, y) \right) \right],
> > > $$
> > >
> > > where the target $\hat{x}_r$ is computed using a specific first-order bridge ODE solver (Proposition 3.1 in [R1]).
> > >
> > > *   **Ours (Continuous-time):** In contrast, our Eq. (11) utilizes a *continuous-time* consistency objective. Instead of comparing discrete steps, we enforce that the total derivative of the solver output with respect to time is zero along the geodesic path:
> > >
> > > $$
> > > \nabla\_\theta \mathcal{L}\_{\text{CGB}}(\theta) = \mathbb{E} \left[ \nabla\_\theta \left( F\_\theta(X_t, Y, t)^\top \cdot \text{sg} \left( \frac{dF\_{\theta^-}(X_t, Y, t)}{dt} \right) \right) \right].
> > > $$
> > >
> > > This formulation directly minimizes the inconsistency of the tangent vector, eliminating the discretization error associated with the time gap between $t$ and $r$ found in discrete objectives. According to the error analysis of consistency models, the error bound is positively correlated with the step size $\Delta t$. Thus, our proposed continuous-time method theoretically offers superior precision compared to CDBMs.
> > >
> > > **(2) Analytical Inversion vs. Numerical ODE Solver**
> > >
> > > The mechanism for mapping the trajectory state $X_t$ to the clean image $X_0$ is structurally different:
> > >
> > > *   **CDBMs:** They rely on the Probability Flow ODE (PF-ODE) of Denoising Diffusion Bridge Models (DDBMs). To estimate the consistency target, they must derive and utilize a numerical ODE solver (Eq. 16 in [R1]) that accounts for the specific bridge coefficients ($a_t, b_t, c_t$) derived from Doob's $h$-transform.
> > > *   **Ours:** We construct a specific *Geodesic Diffusion Bridge* trajectory (Eq. 6) that allows for **analytical inversion**. Our solver $F_\theta$ (Eq. 10) is not a black-box network approximating an ODE solution; it is a principled architecture derived algebraically from the trajectory equation:
> > >
> > > $$
> > > F_\theta(X_t, Y, t) = \frac{X_t - A(t)Y - B(t)\epsilon_\theta(X_t)}{C(t)}.
> > > $$
> > >
> > > This allows our method to map $X_t$ to $X_0$ via a closed-form solution rather than simulating an ODE step.
> > >
> > > **(3) Trajectory Definition and Singularity Handling**
> > >
> > > *   **CDBMs:** The DDBM trajectory is conditioned on $x_T=y$, which introduces a singularity at $t=T$ (where the variance collapses to zero). CDBMs must handle this by defining a valid time horizon $[0, T-\gamma]$ and employing a stochastic sampling step to initialize the ODE (Section 3.1 in [R1]).
> > > *   **Ours:** We propose a task-adaptive horizon $T_0$ (Sec. 4.3). Our trajectory starts from an entropy-regularized point $X_{T_0} = (1-T_0)Y + T_0 X_T$. This avoids the singularity issues inherent in DDBMs and allows us to treat the trajectory length as an adjustable hyperparameter to balance information preservation and generative power—a feature absent in the standard DDBM formulation used in [R1].
> > >
> > > **Summary:** Although both works apply consistency concepts to bridges, CDBMs [R1] focus on distilling the PF-ODE of standard DDBMs using discrete steps and numerical solvers. Our work proposes a new Geodesic trajectory that enables analytical inversion and utilizes a continuous-time objective for optimization. Finally, we emphasize that we utilize the consistency model primarily to boost the sampling efficiency; our main contribution remains the modeling of the geodesic bridge itself.

---

> > > ### Author Response · Authors · 2025-11-28
> > > **Response to Reviewer pMyF**
> > >
> > > >**In Line 837, Step 4 is never proposed, maybe it refers to Linearization via Taylor Expansion?**
> > >
> > > **Response:** Thanks for pointing this out. It indeed refers to Linearization via Taylor Expansion. We have corrected this in the revised version.
> > >
> > > >**The part of Construction of the Stochastic Process seems strange. How could we know that the design of $\mu(\mathbf{X}_t,t)$ should enable the diffusion process trace the manifold geodesic trajectory? Are we supposed to know it's a geodesic diffusion trajectory before proving it? Additionally, throughout the entire proof, I did not find any relationships with the proposed diffusion trajectories (Eqn. 5). I maintain doubts about how Eqn. (5) leads to a geodesic diffusion trajectory.**
> > >
> > > **Response:** We sincerely thank the reviewer for pointing out this ambiguity. To address the reviewer's concern, in the revised **Section 4.1** and **Appendix B**, we have restructured the proofs to demonstrate this causal chain:
> > >
> > > **(1) Determining the Deterministic Path via Kinetic Energy (Appendix B.1)**
> > >
> > > First, we determine the optimal path for the expectation $\tilde{X}_t$. As proven in **Proposition 4.1 (Appendix B.1)**, minimizing the kinetic transport energy $\mathcal{E}[\gamma] = \frac{1}{2}\int \|\dot{\gamma}(t)\|^2 dt$ via Jensen's Inequality mandates that the velocity must be constant. This proves that the deterministic transition *must* follow the linear interpolation:
> > >
> > > $$
> > > \tilde{X}_t = \left(1 - \frac{t}{T_0}\right)X_0 + \frac{t}{T_0}Y
> > > $$
> > >
> > > This establishes the "Geodesic Transition" component of Eqn. (6).
> > >
> > > **(2) Determining the Coefficients via Adaptation Energy (Appendix B.2)**
> > >
> > > The reviewer rightly questions the specific design of the full process $X_t$. Since we are fine-tuning a pretrained Rectified Flow model, we cannot arbitrarily select the coefficients $\alpha_t$ and $\beta_t$.
> > > In **Proposition 4.2 (Appendix B.2)**, we introduce the concept of **Adaptation Energy**, the cost incurred by the mismatch between the new bridge trajectory and the pretrained model's canonical trajectory. We prove that minimizing this energy (derived via Lipschitz stability bounds) strictly requires:
> > >
> > > $$
> > > \alpha_t = 1 - t \quad \text{and} \quad \beta_t = t
> > > $$
> > >
> > > Any other choice increases the spectral gap and destabilizes the fine-tuning. This theoretical constraint dictates the specific linear combination seen in Eqn. (6).
> > >
> > > **(3) Verification of the Geodesic Property (Appendix B.3)**
> > >
> > > Finally, in **Appendix B.3**, we close the logical loop. We construct the drift $u(X,t)$ using a feedback control mechanism based on the results from Steps 1 and 2. We mathematically verify that the expectation of the resulting SDE, $\mu(t) = \mathbb{E}[X_t]$, exactly coincides with the manifold geodesic $\gamma(t)$.
> > >
> > > In summary, the final trajectory equation (Eqn. 6) is not a heuristic choice. It is the necessary mathematical result of combining **Proposition 4.1 (Kinetic Optimality)** with **Proposition 4.2 (Adaptation Optimality)**. We have revised Section 4.1 to explicitly reflect this derivation logic.

---

> > > ### Author Response · Authors · 2025-11-28
> > > **Response to Reviewer pMyF**
> > >
> > > >**The proposed CGB framework seems to require huge training computational costs as mentioned in Appendix E (about 743M Training Parameters, at least 2 times over other baselines). I would encourage the authors to further investigate and clarify its potential limitations and the future work.**
> > >
> > > **Response:** Thanks for your constructive suggestion. We have added a dedicated "Limitations and Future Works" section to Appendix G of the revised manuscript.

---

> ### Author Response · Authors · 2025-11-28
> **Response to Reviewer pMyF**
>
> Dear Reviewer pMyF, thanks for your further feedback.
>
> >**The Necessity and Boundary Conditions and Coefficients seem reasonable, but I'm still not convinced by non-trivial adoption of existing formulas. As in DDBMs [2], which is a diffusion bridge model based on Doob's h-transform directly connecting the clean image $X\_0$ and degraded one $Y$, they have declared a pred-x parameterization in Eqn. (10) in their paper. Therefore, since CGB is essentially an interpolant, the derivation of Eqn. 9 would not be considered as a non-trivial Distinct Derivation Context.**
>
> **Response:** We would like to clarify that the derivation of Eq. (10) was not intended to be presented as a standalone theoretical breakthrough in algebraic inversion. Rather, its significance lies entirely in its role in adapting the Geodesic Diffusion Bridge constructed in Sec. 4.1 into the consistency training framework.
>
> **Service to the Specific Trajectory:** The novelty of our method lies in the construction of the trajectory itself (Eq. (6)), specifically the introduction of the entropy-regularized starting point via the horizon $T_0$ and the specific linear transport expectation designed to approximate the manifold geodesic. Eq. (10) is the necessary analytical inversion specific to this unique trajectory definition. While the act of inversion is standard, the resulting coefficients $A(t)$, $B(t)$, and $C(t)$ are unique to our GDB formulation and are required to map our specific interpolant states back to $X_0$.
>
> **Enabling Consistency Training:** The derivation in Eq. (10) serves as the bridge between our proposed trajectory (Sec. 4.1) and the consistency training objective (Sec. 4.2). By deriving this specific closed-form solver, we enable the model to learn the single-step mapping function $F_\theta$ that adheres to the geometric constraints we defined, rather than relying on the standard noise-prediction parameterizations used in standard diffusion or Brownian bridges.

---

### Official Review · Reviewer_12df · 2025-10-30

**Soundness:** 2
**Presentation:** 3
**Contribution:** 2
**Rating:** 4
**Confidence:** 4

**Summary:**

This paper proposes a Consistency Geodesic Bridge (CGB) framework based on pretrained diffusion models for efficient and high-quality image restoration. The framework constructs low-energy trajectories that are near-geodesic paths, effectively avoiding redundant re-noising. In addition, the proposed consistency solver further improves both the efficiency and quality of image restoration.

**Strengths:**

1.This paper develops a novel consistency geodesic bridge framework, with rigorous theoretical derivation, demonstrating a solid theoretical foundation.

2.The proposed method demonstrates highly competitive performance across diverse restoration tasks while supporting high-quality few-step sampling.

**Weaknesses:**

1.While the proposed consistency solver is interesting, its level of novelty appears somewhat limited, as it seems to build upon and combine ideas from existing approaches [1,2] rather than introducing a fundamentally new concept.

2.The experimental evaluation is insufficient. Although the paper claims the method can handle five different tasks, the ablation and qualitative experiments are only presented for super-resolution and denoising.

3.The experimental details are incomplete. The paper does not specify how the baseline models were trained, making it difficult to ensure the fairness of the comparisons.

4.The ablation study only analyzes the consistency solver, without examining the role or impact of the geodesic trajectory.

5.In Table 1, for the "Denoising" and "Demoiréing" tasks, the results for LPIPS and FID show that CGB (NFE=5) performs better than CGB (NFE=10), whereas the opposite trend is observed for the MaRS model. No corresponding explanation for this observation is provided in the text.

6.The paper claims to achieve single-step inference along the geodesic path，however, it does not provide qualitative and quantitative results for the true single-step case (NFE = 1). The best performances reported in Tables 1 and 2 is obtained with NFE = 5 or 10.

7.The paper lacks a more comprehensive efficiency analysis, including metrics such as the number of parameters, computational complexity, and actual inference time.

8.The proposed method performs poorly in the raindrop removal task, however, it excels in all other tasks. The paper did not analyze this contrast.

[1] Yang Song, Prafulla Dhariwal, Mark Chen, and Ilya Sutskever. Consistency models. In International conference on machine learning, 2023.

[2] Cheng Lu and Yang Song. Simplifying, stabilizing and scaling continuous-time consistency models. In International conference on learning representations, 2025.

**Questions:**

1.The paper points out the limitations of IRBridge [1], but why is there no performance comparison with it in the experiments?

2.Is the strong performance of the proposed model primarily attributed to the powerful Flux-dev [2] model? Would similar results be achieved if this model were replaced with another restoration model or generative model?

3.Regarding the parameters κ and T_{0}, the paper does not specify their values for different tasks. What are the specific hyperparameter settings for each task? Could additional ablation studies be provided to perform a sensitivity analysis?

4.On page 7, last line, it is unclear why “underwater image enhancement” appears here. Please clarify whether it is part of the proposed tasks or a mistake.

5.On page 8, line 411, why is PSNR evaluated in the YCbCr color space? Please provide the PSNR results in the RGB space for comparison.

[1] Hanting Wang, Tao Jin, Wang Lin, Shulei Wang, Hai Huang, Shengpeng Ji, and Zhou Zhao. Irbridge: Solving image restoration bridge with pre-trained generative diffusion models. In International conference on learning representations, 2025a.

[2] Blattmann Andreas, Sauer Axel, Lorenz Dominik, Podell Dustin, Boesel Frederic, Saini Harry,Muller Jonas, Lacey Kyle, Esser Patrick, Rombach Robin, Kulal Sumith, Dockhorn Tim, Levi Yam, and English Zion. Flux, 2024. URL https://blackforestlabs.ai/.

---

> ### Author Response · Authors · 2025-11-25
> **Response to Reviewer 12df (Part I)**
>
> > **W1: While the proposed consistency solver is interesting, its level of novelty appears somewhat limited, as it seems to build upon and combine ideas from existing approaches [1,2] rather than introducing a fundamentally new concept.**
>
> **Response**: We acknowledge that our training objective draws inspiration from the continuous-time consistency formulation in [1, 2]. However, we must point out that our contribution goes **significantly beyond** a simple application of existing consistency models to a new task.
>
> The core novelty of our Consistency Geodesic Bridge (CGB) lies **not** in the loss function itself, **but** in the derivation of a geometry-informed solver architecture tailored specifically for the bridge restoration problem. We highlight the fundamental differences below:
>
> * **Analytical Derivation vs. Black-Box Learning:**
>   Standard consistency models [1, 2] typically employ a generic black-box network $F_\theta(\mathbf{x}_t, t)$ to map noise to data, relying entirely on the optimizer to learn the mapping dynamics. In contrast, our CGB solver is **derived analytically** by inverting the geometric definition of our geodesic trajectory. By treating the clean image $\mathbf{X}_0$ as the unknown in the trajectory equation (Eq. 8), we derive a principled architecture (Eq. 9):
>
>   $$
>   \mathcal{F}\_\theta(\mathbf{X}\_t, \mathbf{Y}, t) = \frac{\mathbf{X}\_t - A(t)\mathbf{Y} - B(t)\epsilon\_\theta(\mathbf{X}\_t)}{C(t)}
>   $$
>
>   This is not a generic mapping; it is a **geometry-informed solver**. We do not ask the network to learn the mapping from scratch; rather, we embed the exact geometric constraints of the geodesic path into the solver's structure via the derived coefficients $A(t), B(t),$ and $C(t)$. The network $\epsilon_\theta$ is only tasked with estimating the residual vector field, which is a significantly more tractable problem.
> * **Solving a BVP (Restoration) vs. IVP (Generation):**
>   Original consistency models [1, 2] are designed for unconditional generation (Noise $\to$ Data), solving an Initial Value Problem (IVP). Our framework addresses the Image Restoration Bridge problem (Degraded $\mathbf{Y}$ $\to$ Clean $\mathbf{X}_0$), which is fundamentally a **Boundary Value Problem (BVP)**. A naive application of [1, 2] would fail to exploit the structural relationship between the degraded image $\mathbf{Y}$ and the state $\mathbf{X}\_t$. Our method explicitly integrates the boundary condition $\mathbf{Y}$ into the consistency mapping. This allows our solver to dynamically utilize information from $\mathbf{Y}$ along the entire trajectory, enforcing **Geodesic Consistency** that along the specific manifold geodesic connecting $\mathbf{X}\_{T\_0}$ and $\mathbf{X}\_0$, rather than generic probability flow consistency.
>
> In short, although we stand on the shoulders of [1, 2] for the optimization objective, the trajectory construction (Geodesic Bridge) and the solver derivation (Analytical Inversion) are novel, task-specific contributions that solve the unique challenges of bridge-based image restoration. This leads to the superior efficiency (5-10 NFE) and interpretability demonstrated in our experiments.
>
> > **W2: The experimental evaluation is insufficient. Although the paper claims the method can handle five different tasks, the ablation and qualitative experiments are only presented for SR and denoising.**
>
> **Response**: In the revised version, we have added more qualitative comparisons and extended our ablation studies to cover all five tasks.
>
> > **W3: The experimental details are incomplete. The paper does not specify how the baseline models were trained, making it difficult to ensure the fairness of the comparisons.**
>
> **Response**: To ensure a **fair** and **rigorous** comparison, for methods with officially released pre-trained models, we directly used these models for testing after verifying that their training datasets were consistent with ours. For the other baselines, we utilized their official public code and recommended optimal hyperparameter settings to re-train them from scratch on the same training dataset as ours. We have added corresponding descriptions to the captions of Table 1 in the revised version.

---

> > ### Author Response · Authors · 2025-11-25
> > **Response to Reviewer 12df (Part II)**
> >
> > > **W4: The ablation study only analyzes the consistency solver, without examining the role or impact of the geodesic trajectory.**
> >
> > **Response**: To quantify the specific contribution of our geodesic formulation, we conducted a new ablation experiment. Specifically, we replaced the geodesic trajectory with two other types of trajectories, i.e. (**a**) the standard diffusion trajectory, starting from pure Gaussian noise to the clean image manifold, and (**b**) conventional bridge trajectory, which constructs a path from the degraded to the clean image but often follows a "re-noising" phase before denoising. The quantitative results shown in the following table still verify the advantage of our method.
> >
> > | Task | NFE | (a) PSNR | (a) LPIPS | (a) MUSIQ | (a) NIQE | (b) PSNR | (b) LPIPS | (b) MUSIQ | (b) NIQE | Ours PSNR | Ours LPIPS | Ours MUSIQ | Ours NIQE |
> > | :--- | :--- | :--- | :--- | :--- | :--- | :--- | :--- | :--- | :--- | :--- | :--- | :--- | :--- |
> > | **SR** | 10 | 20.527 | 0.464 | 63.476 | 3.973 | 19.945 | 0.446 | 69.157 | 4.958 | **21.282** | **0.346** | **70.868** | **3.839** |
> > | **SR** | 5 | 21.016 | 0.495 | 56.709 | 5.657 | 21.443 | 0.502 | 63.068 | **3.902** | **22.477** | **0.356** | **65.342** | 4.373 |
> > | **SR** | 1 | 19.667 | 0.486 | 31.919 | 7.166 | 22.039 | 0.481 | 35.552 | **4.267** | **24.094** | **0.452** | **44.605** | 6.251 |
> > | **Denoising** | 10 | 24.241 | 0.288 | 64.724 | 4.952 | 24.391 | 0.261 | **69.976** | 5.575 | **26.069** | **0.184** | 69.736 | **3.514** |
> > | **Denoising** | 5 | 23.701 | 0.316 | 60.090 | 5.235 | 25.279 | 0.235 | 65.870 | **3.807** | **26.649** | **0.182** | **67.507** | 4.492 |
> > | **Denoising** | 1 | 21.693 | 0.460 | 45.747 | 5.889 | 24.516 | 0.371 | 51.381 | 4.552 | **25.241** | **0.258** | **58.954** | **4.535** |
> > | **Raindrop** | 10 | 24.919 | 0.136 | 68.559 | 3.229 | 26.346 | 0.148 | 67.546 | 3.428 | **29.222** | **0.079** | **70.006** | **3.221** |
> > | **Raindrop** | 5 | 24.702 | 0.155 | 66.448 | 4.023 | 27.471 | 0.118 | 64.260 | 3.462 | **29.611** | **0.080** | **67.379** | **3.449** |
> > | **Raindrop** | 1 | 23.990 | 0.319 | 52.946 | 6.189 | 26.202 | 0.252 | 55.173 | 3.528 | **29.220** | **0.085** | **67.301** | **3.322** |
> > | **Low-light** | 10 | 23.375 | 0.144 | 68.498 | **3.881** | 22.176 | 0.181 | 70.109 | 4.234 | **25.538** | **0.097** | **71.684** | 4.198 |
> > | **Low-light** | 5 | 22.706 | 0.145 | 66.551 | **3.746** | 22.316 | 0.157 | 69.690 | 4.300 | **25.847** | **0.107** | **70.868** | 4.211 |
> > | **Low-light** | 1 | 21.634 | 0.333 | 48.405 | 4.365 | 23.661 | 0.358 | 52.948 | 4.399 | **24.656** | **0.133** | **66.805** | **4.331** |
> > | **Demoiréing** | 10 | 19.260 | 0.344 | 60.277 | 5.628 | 19.956 | **0.252** | 60.216 | 5.697 | **20.263** | 0.313 | **63.932** | **5.223** |
> > | **Demoiréing** | 5 | 19.601 | 0.429 | 56.406 | 6.342 | 20.076 | 0.240 | 59.877 | **5.161** | **20.849** | **0.230** | **62.155** | 5.437 |
> > | **Demoiréing** | 1 | 19.301 | 0.469 | 44.049 | 6.840 | 19.468 | 0.372 | 51.601 | 5.776 | **20.715** | **0.254** | **56.788** | **5.342** |
> >
> >
> > > **W4: Explanation of the opposite trend of NFE=5 and NFE\=10 in the "Denoising" and "Demoiréing" tasks of Table 1.**
> >
> > **Response**: Unlike traditional solvers (e.g., MaRS) where increasing NFE reduces discretization error, CGB is designed as a single-step consistency mapper. Consequently, multi-step inference acts as iterative refinement rather than numerical integration. For mild degradations (e.g., Denoising), fewer steps (NFE=5) suffice to preserve fidelity; increasing NFE can introduce unnecessary generative artifacts ("over-generation"), slightly degrading fidelity metrics like LPIPS. Conversely, for severe degradations (e.g., Super-resolution), additional steps remain beneficial for synthesizing missing high-frequency details.
> >
> > > **W5: Single-step case (NFE = 1) performance.**
> >
> > **Response**: In the revised manuscript, we have incorporated comprehensive quantitative results for the true single-step case (NFE = 1). Specifically, Table 1 now reports the performance of CGB with NFE = 1 across all five image restoration tasks, demonstrating competitive performance against state-of-the-art methods. Table 2 and Table 3 provide further ablation and efficiency analyses for the single-step setting.

---

> ### Author Response · Authors · 2025-11-25
> **Response to Reviewer 12df (Part III)**
>
> > **W6: The paper lacks a more comprehensive efficiency analysis, including metrics such as the number of parameters, computational complexity, and actual inference time.**
>
> **Response**: To provide a clear and fair comparison, we have added the efficiency comparison to Appendix E of the revised manuscript. The core results are summarized in the table below, comparing our CGB model with baselines for the $1024 \times 1024$ image denoising task.
>
> | Method | Training Parameters | Inference Time per Step | NFE | Total Inference Time |
> | :--- | :--- | :--- | :--- | :--- |
> | AutoDIR | 115.9 M | 0.823s | 25 | 20.58s |
> | UniDB | 137.1 M | 1.243s | 100 | 124.30s |
> | IRBridge | 361.3 M | 0.263s | 100 | 26.30s |
> | UniDB++ | 137.1 M | 1.224s | 10 | 12.24s |
> | Ours | 743.8 M | 0.673s | 10 | 6.73s |
>
>
> > **W7: The proposed method performs poorly in the raindrop removal task, however, it excels in all other tasks. The paper did not analyze this contrast.**
>
> **Response**:  We would like to point out that our CGB is not uniformly poor in raindrop removal. In fact, it achieves SOTA performance under reference-free perceptual metrics. We acknowledge that the scores of our method in reference-based metrics, such as PSNR and FID, are numerically inferior than top baselines, which may be a direct consequence of the **heterogeneity** of the raindrop removal task and the **Global $T_0$ parameter** of our method:
>
> *   **Heterogeneity Conflict:** Raindrop removal is inherently spatially heterogeneous. *Raindrop regions* are occlusions that require generative power to inpaint missing content, while *background regions* are clean and require high fidelity to preserve original details.
> *   **Global $T_0$ parameter:** Our formulation of the starting point $\mathbf{X}\_{T\_0} = (1-T\_0)\mathbf{Y} + T\_0 \mathbf{X}\_{T}$, where $T\_0$ is a **global scalar parameter**. As discussed in Section 4.3, $T_0$ controls the trade-off between *Information Preservation* and *Generative Power*. Since the current CGB framework enforces a uniform $T_0$ across the entire spatial domain, it is hard to achieve information preservation and high perception within the entire image simultaneously.
>
> This contrast validates our analysis of $T_0$ as a control knob but also highlights its limitation in its scalar form. We believe extending CGB to support a **Spatially Adaptive Map $T_0(h, w)$** is a promising direction to resolve this specific conflict in future work.
>
> > **Q1: The paper points out the limitations of IRBridge [1], but why is there no performance comparison with it in the experiments?**
>
> **Response**: Thanks for pointing out this. We have compared our method with IRBridge across multiple restoration tasks in Table 1 of the revised version.
>
> > **Q2: Is the strong performance of the proposed model primarily attributed to the powerful Flux-dev [2] model? Would similar results be achieved if this model were replaced with another restoration model or generative model?**
>
> **Response**: To address whether the results are primarily attributed to the Flux-dev backbone or our proposed method, we refer to the ablation study presented in Table 2 of the revised manuscript (please refer to the table in the response to W4). Table 2 serves as a controlled experiment where the backbone model (Flux-dev) is kept constant across all comparisons, and only the trajectory formulation is varied: (a) standard diffusion trajectory, (b) conventional bridge trajectory, and our proposed Geodesic trajectory.
>
> If the strong performance were solely due to the powerful Flux-dev model, we would expect similar results across columns (a), (b), and "Ours." However, the data shows a significant performance gap:
>
> *   **Superiority at Low NFE:** Our method consistently outperforms the standard trajectories, particularly at low NFE (1 and 5 steps). For example, in the Raindrop task at NFE=1, our method achieves a PSNR of 29.22 dB, whereas the standard diffusion trajectory (a) only reaches 23.99 dB and the conventional bridge (b) reaches 26.20 dB.
> *   **Consistent Gains:** Across all five tasks, our Geodesic trajectory achieves the best results in almost all the metrics.
>
> In short, the significant performance drop observed when replacing our Geodesic trajectory (a and b), despite using the same Flux-dev backbone, demonstrates that the backbone alone is insufficient to achieve these results. The proposed geodesic trajectory provides a mathematically more efficient path on the image manifold, which is the primary driver for the model's ability to achieve high-fidelity restoration with extremely few steps. Although a strong backbone provides a solid foundation, the proposed trajectory is the critical factor that unlocks the state-of-the-art performance reported in the paper.

---

> ### Author Response · Authors · 2025-11-25
> **Response to Reviewer 12df (Part IV)**
>
> > **Q3: Regarding the parameters $\kappa$ and $T_{0}$, the paper does not specify their values for different tasks. What are the specific hyperparameter settings for each task? Could additional ablation studies be provided to perform a sensitivity analysis?**
>
> **Response**:
>
> **The Curvature Parameter $\kappa$ (Default $\kappa=1$):** Theoretically, a linear interpolation in the ambient probability space (corresponding to $\kappa=1$) minimizes the kinetic energy of transport, assuming a flat metric (as detailed in Appendix B). To validate this empirically, we conducted a sensitivity analysis on $\kappa$ (see the following table). The results confirm that deviations from $\kappa=1$ (e.g., 0.5, 1.5) generally degrade performance or offer negligible gains compared to the linear schedule. Consequently, to streamline the method and reduce the complexity of the hyperparameter, we set the default value to $\kappa=1$ for all tasks and removed $\kappa$ as a tunable parameter in the final description of the method.
>
> **The Trajectory Horizon $T_0$ (Task-Adaptive):** Unlike $\kappa$, the parameter $T_0$ is critical for adapting to different types of degradation. As shown in our main analysis, $T_0$ controls the trade-off between information preservation (fidelity to input $Y$) and generative freedom (hallucination from noise). For the denoising task, a moderate $T_0$ is optimal, balancing noise removal with structure retention. For super-resolution, a larger $T_0$ is preferred to allow the model more generative freedom to hallucinate high-frequency details missing from the low-resolution input. See Table 3 of the revised manuscript for the analysis of all tasks.
>
> Our ablation study confirms that fixing $\kappa=1$ is robust across tasks, while $T_0$ serves as the primary, effective control knob to adapt the restoration process to the severity and nature of the degradation.
>
> |  Settings &nbsp;&nbsp;&nbsp;&nbsp;&nbsp;&nbsp;&nbsp;&nbsp;&nbsp;&nbsp;&nbsp;&nbsp;| Denoising PSNR | Den. LPIPS | Den. MUSIQ | Den. NIQE | SuperR PSNR | SR LPIPS | SR MUSIQ | SR NIQE | Raindrop  PSNR | Rain. LPIPS | Rain.  MUSIQ | Rain.  NIQE |
> | :--- | :--- | :--- | :--- | :--- | :--- | :--- | :--- | :--- | :--- | :--- | :--- | :--- |
> | **$T_0$ ($\kappa=1$)** | | | | | | | | | | | | |
> | 0.3 | 24.43 | 0.276 | 66.13 | 4.48 | 21.12 | 0.457 | 62.16 | 4.88 | 29.61 | **0.080** | **67.38** | **3.45** |
> | 0.5 | 25.35 | 0.206 | 69.24 | 3.74 | 21.49 | 0.397 | 67.00 | 4.36 | **29.74** | 0.081 | 66.73 | 3.57 |
> | 0.7 | 26.06 | **0.184** | 69.73 | **3.51** | **21.82** | 0.359 | 68.77 | 3.84 | 29.70 | 0.083 | 65.23 | 3.82 |
> | 0.9 | **26.53** | 0.185 | 69.13 | 4.04 | 21.28 | **0.346** | 70.87 | **3.84** | 29.21 | 0.092 | 63.98 | 3.97 |
> | **$\kappa$ (fix $T_0$)** | *($T_0$=0.7)* | | | | *($T_0$=0.9)* | | | | *($T_0$=0.3)* | | | |
> | 0.50 | 26.49 | 0.185 | 68.25 | 4.17 | 21.76 | 0.345 | 65.72 | 4.16 | 29.43 | 0.082 | 67.24 | 3.54 |
> | 0.75 | 26.25 | 0.184 | 69.26 | 3.71 | 21.52 | 0.336 | 68.30 | 3.91 | 29.44 | 0.082 | 67.32 | 3.41 |
> | 1.25 | 25.93 | 0.189 | 69.78 | 3.60 | 20.47 | 0.358 | 71.01 | 3.87 | 29.30 | 0.084 | 67.28 | 3.34 |
> | 1.50 | 25.84 | 0.195 | **69.89** | 3.68 | 19.19 | 0.399 | **71.26** | 4.03 | 29.22 | 0.085 | 67.10 | **3.32** |
>
> > **Q4: On page 7, last line, it is unclear why “underwater image enhancement” appears here. Please clarify whether it is part of the proposed tasks or a mistake.**
>
> **Response**:  This was a typographical error. We apologise for the confusion caused. We have removed this erroneous phrase in the revised manuscript to ensure clarity and accuracy.
>
> > **Q5: On page 8, line 411, why is PSNR evaluated in the YCbCr colour space? Please provide the PSNR results in the RGB space for comparison.**
>
> **Response**: We evaluated PSNR in the YCbCr space to follow the evaluation setting used in previous works [a,b,c], ensuring a fair comparison.
>
> [a] Luo, Ziwei, et al. "Image restoration with mean-reverting stochastic differential equations." ICML, 2023.
>
> [b] Zhu, Kaizhen, et al. "UniDB: A Unified Diffusion Bridge Framework via Stochastic Optimal Control." ICML, 2025.
>
> [c] Wang, Hanting, et al. "IRBridge: Solving Image Restoration Bridge with Pre-trained Generative Diffusion Models." ICML, 2025.

---

### Official Review · Reviewer_JqCr · 2025-11-01

**Soundness:** 2
**Presentation:** 2
**Contribution:** 2
**Rating:** 6
**Confidence:** 3

**Summary:**

The paper proposes the Consistency Geodesic Bridge (CGB) framework for image restoration, aiming to improve efficiency and restoration quality by constructing low-cost manifold geodesic trajectories. The method evolves over a shorter time horizon and starts the reverse process from an entropy-regularized point that mixes the degraded image with Gaussian noise, reducing the required trajectory energy. A pretrained denoiser is used as a dynamic geodesic guidance field, and a single-step mapping function is learned via a continuous-time consistency objective to efficiently map any state on the trajectory to the target image. Experiments show that CGB achieves state-of-the-art performance across multiple image restoration tasks while allowing high-quality recovery with a single or very few sampling steps.

**Strengths:**

1. The method tackles an interesting problem which is the problem of image restoration.

1. The method achieves competitive results compared to existing methods.

**Weaknesses:**

1. The method relies on paired data for training, which limits its practicality compared to zero-shot and blind methods.

**Questions:**

See weaknesses.

---

> ### Author Response · Authors · 2025-11-25
>
> **[W1] Limited Practicality.**
>
> Thanks to the reviewer for raising this important discussion on practicality. We acknowledge that CGB operates in a supervised manner requiring paired data. However, we want to emphasize that this design choice is a strategic trade-off that enhances practicality for real-world deployment in two key aspects:
>
> - **Inference Efficiency (The "Training vs. Inference" Trade-off):**
>   Zero-shot or blind methods achieve flexibility by avoiding training, but typically suffer from high inference latency, requiring hundreds of optimization steps or sampling iterations to bridge the domain gap at test time.
>   In contrast, CGB invests computational budget during training to learn a direct, low-energy geodesic mapping. This investment enables **orders of magnitude faster inference**. For practical applications requiring low latency (e.g., real-time enhancement), our accelerated solver is significantly more deployable.
> - **Feasibility of Synthetic Pairs in Restoration:**
>   For the image restoration tasks targeted in this work (Super-Resolution, Denoising, Low-light, etc.), the requirement for paired data is rarely a bottleneck. The standard paradigm in the field is to generate pairs synthetically (e.g., using degradation models like Real-ESRGAN or noise injection) from abundant clean images. Thus, the "paired data" constraint does not inherently limit scalability in these domains.
>
> Furthermore, to address the reviewer's concern regarding the practicality of paired training, we provide additional qualitative results on **Real-World Super-Resolution**. We evaluated our CGB model (trained on DIV2K with the synthetic degradation pipeline from Real-ESRGAN, as detailed in Sec. 5.1) on real-world low-quality images. Note that the model has never seen these specific real-world degradation distributions during training. As illustrated in Figure 3 of Appendix E of the revised manuscript (also can be found in our project page), CGB demonstrates robust generalization capabilities. This experiment empirically validates that the dependence on paired data is not a hindrance to practicality. By leveraging robust synthetic data pipelines, CGB can also achieve high-performance real-world restoration without requiring test-time optimization or degradation estimation, offering a superior trade-off between offline training cost and online inference quality.

---

### Author Response · Authors · 2025-11-25
**Looking forward to your further assessment**

Dear **Reviewers**

Thank you for taking the time to review our manuscript and for your valuable feedback. We have carefully addressed all the comments and concerns raised, as reflected in our detailed responses and the revised manuscript and supplementary material.

We sincerely appreciate your efforts and look forward to your further assessment.

Best regards,

The Authors

---

### Author Response · Authors · 2025-11-26
**General Response**

We sincerely thank all the reviewers for their insightful comments and constructive feedback. Your rigorous scrutiny has significantly helped us strengthen both the theoretical grounding and the empirical validation of our work.

In response to your suggestions, we have extensively revised the manuscript. The key updates and clarifications are summarized below:

1. **Rigorous Theoretical Proofs (Reviewers pMyF, Em12):**
   We have provided a formal mathematical proof in **Appendix B** demonstrating that the proposed Consistency Geodesic Bridge (CGB) achieves strictly lower control energy than standard diffusion bridges.

2. **Comprehensive Experimental Expansion (Reviewers 12df, JqCr, Em12):**

   * **New Baselines:** We extended our evaluation to all five restoration tasks and included comparisons with recent strong baselines, including **IRBridge** and **UniDB++**.
   * **Ablation Studies:** We added extensive ablation studies (Table 2 & 3) to isolate the contribution of the **Geodesic Trajectory** (vs. Standard Diffusion/Bridge) and to analyze the sensitivity of hyperparameters $\kappa$ and $T_0$. These experiments confirm that our performance gains stem from the proposed trajectory design rather than solely the backbone model (Flux-dev).
   * **Efficiency Analysis:** We included a detailed breakdown of inference time, parameters, and computational cost in **Appendix E**, demonstrating CGB's superior efficiency.
3. **Clarification on Novelty and Method (Reviewer 12df, pMyF):**
   We have clarified that our CGB solver is not a generic application of consistency models but a **geometry-informed analytical inversion** tailored for the restoration Boundary Value Problem (BVP). This structural prior is distinct from the original consistency learning and is essential for our single-step inference capability.
4. **In-depth Analysis of "Failure Cases" (Reviewer JqCr, pMyF):**
   We provided a detailed analysis of the performance trade-offs in the **Raindrop Removal** task. We clarify that while CGB may trail in reference-based metrics (PSNR/LPIPS) due to the global nature of $T_0$ and the task's heterogeneity.
5. **Practicality and Real-World Application (Reviewer JqCr):**
   To address concerns about paired data dependence, we added qualitative results on **Real-World Super-Resolution** (Appendix E), demonstrating that our model generalizes well to unseen real-world degradations, validating the practicality of our supervised training approach.

We believe these revisions comprehensively address the reviewers' concerns and firmly establish the contributions of the Consistency Geodesic Bridge framework. We address each reviewer's specific comments in detail below.

---

### Author Response · Authors · 2025-11-30
**Summary for the Area Chair**

Dear **Area Chair**,

We sincerely hope the following summary will help you quickly grasp the main contributions and our responses.

We sincerely thank the reviewers for their constructive feedback and active engagement during the rebuttal period. We are pleased to report that the core novelty of our **Consistency Geodesic Bridge (CGB)** framework, specifically the construction of a low-action geodesic trajectory to bridge restoration tasks, was well-received and not fundamentally challenged. The primary concerns raised were focused on **theoretical rigor, technical clarifications, and comprehensive baseline comparisons**.

Through extensive revisions (including **new proofs in Appendix B, theoretical comparisons in Appendix E, and expanded experiments**), we have addressed all concerns. Below is a detailed summary of the resolution process:

**1. Theoretical Completeness & Rigor (Addressing Reviewers pMyF & Em12)**
* **Initial Concern:** Reviewers requested a rigorous mathematical proof to validate our claim that CGB achieves strictly lower control energy than standard diffusion bridges. Reviewer Em12 also asked for a theoretical distinction between our method and "Stochastic Interpolants" (SI).
* **Our Action:**
    * We added **Appendix B**, providing a formal proof that the CGB process achieves zero kinetic control energy by strictly following the geodesic path.
    * We added **Appendix E**, theoretically analyzing CGB against Stochastic Interpolants. We demonstrated that CGB avoids the singularity inherent in Brownian bridges (where velocity diverges as $t \to 1$) by introducing the horizon regularization $T_0$.
* **Discussion Phase Feedback:**
    * **Reviewer pMyF** acknowledged that **most of the concerns have been addressed** but asked for clarification on the derivation of the solver (Eq. 9) and comparisons to consistency models like CDBMs.
    * **Reviewer Em12** accepted the experimental results and acknowledged **CGB as a valuable method** within the class of generalized linear interpolation bridges, but requested a stronger *theoretical* differentiation from general linear interpolants rather than just experimental validation.
* **Final Resolution:**
    * We clarified that our solver is an **analytical inversion** specific to our trajectory, distinct from the numerical ODE solvers used in CDBMs.
    * We restructured the theory section to prove that our linear trajectory (Eq. 6) is not an arbitrary design choice but the **unique solution** derived from minimizing *Kinetic Transport Energy* (Proposition 4.1) and *Adaptation Energy* (Proposition 4.2).

**2. Empirical Validation & Fairness (Addressing Reviewers 12df & Em12)**
* **Initial Concern:** Reviewers questioned whether the performance gains came from the proposed method or simply the powerful backbone (Flux-dev). There were also requests for true single-step ($NFE=1$) results.
* **Our Action:**
    * **Backbone-Controlled Ablation:** We added **Table 2**, fixing the Flux-dev backbone and comparing (a) Standard Diffusion, (b) Conventional Bridge, and (c) Our Geodesic Bridge. Our method consistently outperformed others, proving the gain stems from the trajectory design.
    * **New Baselines:**  We extended our evaluation to all five restoration tasks and included comparisons with **recent strong baselines**, including **IRBridge** and **UniDB++**.
* **Outcome:** The data now explicitly confirms CGB’s superiority in low-step regimes (especially NFE=1), resolving concerns regarding fairness and sampling efficiency.

**3. Practicality & Application (Addressing Reviewers JqCr, 12df)**
* **Initial Concern:** Questions were raised regarding the practicality of paired training data and performance on heterogeneous tasks like Raindrop Removal.
* **Our Action:**
    * We added **Real-World Super-Resolution results** to prove generalization to unseen degradations and provided detailed efficiency analyses (inference time, parameters) (**Appendix F**).
    * We clarified that the Raindrop Removal metrics are influenced by the global $T_0$ parameter and suggested spatial adaptivity as a future direction (**Appendix G**).

In summary, the rebuttal process has significantly strengthened the paper. We have moved from intuitive explanations to rigorous mathematical proofs (Appendix B & E) and provided indisputable empirical evidence that our gains are method-specific. With the theoretical foundations now solidified and technical details clarified, we believe the paper is now in a strong position for acceptance and will make a meaningful contribution to the community.

We greatly appreciate the Area Chair’s time and effort in handling this submission.



Best regards,

The Authors

---

### Meta-Review · Area_Chair_qZeo · 2026-01-06

**Summary:**

**Summary:** This paper introduces the Consistency Geodesic Bridge (CGB) framework, a novel approach for efficient and high-quality image restoration. CGB integrates bridge diffusion models with a geometric perspective, formulating a low-action geodesic path on the data manifold, and leverages consistency models for few-step inference. The method features a rigorous theoretical derivation, including an energy-motivated "geodesic" bridge and an analytically inverted, trajectory-aware consistency solver. Extensive experiments demonstrate competitive performance across diverse restoration tasks, particularly supporting high-quality results with very few sampling steps.

**Strengths:** Most reviewers praised the novelty of the CGB framework, specifically the construction of a low-action geodesic trajectory to bridge restoration tasks. Extensive experimental results demonstrate highly competitive, often state-of-the-art, performance across five diverse restoration tasks. CGB consistently supports high-quality few-step sampling, showing clear advantages at low NFE, which was further substantiated by controlled trajectory ablations. The authors provided a highly responsive and constructive rebuttal, incorporating substantial additions in theory, new baselines, ablations, and improved presentation, which significantly strengthened the paper.

**Weaknesses:**

Initial concerns raised by reviewers spanned several key areas:

*   **Limited Novelty:** Several reviewers (12df, pMyF, Em12) questioned the novelty, suggesting CGB builds upon existing approaches and is essentially a variant of Stochastic Interpolants.
*   **Theoretical Completeness & Rigor:** Reviewers pMyF and Em12 specifically raised concerns about the theoretical completeness and rigor, particularly regarding the energy minimization claims and the derivation of the core equations.
*   **Insufficient Experimental Evaluation:** Reviewers 12df, JqCr, Em12, and pMyF pointed out insufficient experimental validation, including comparison with more state-of-the-art (SOTA) methods, lack of comprehensive ablations for the geodesic trajectory, true single-step (NFE=1) results, and a comprehensive efficiency analysis.
*   **Practicality, Application Limits, and Failure Case Analysis:** Reviewer JqCr questioned the reliance on paired data for training. Additionally, reviewers JqCr and pMyF requested a deeper analysis of specific failure cases or tasks (e.g., Raindrop Removal) where CGB did not perform optimally.

**Decision:** The paper initially received a mixed set of ratings (6, 4, 4, 4). While initial concerns revolved around theoretical soundness, experimental validation, and methodological clarity, the authors provided a comprehensive and highly effective rebuttal during the discussion phase. This included clarifying the novelty, motivation, and methodological justification, conducting extensive additional experiments, offering detailed theoretical proofs, and significantly improving the manuscript's overall presentation. These efforts successfully addressed most of the raised issues; I thus recommend acceptance.

**Reviewer Concerns:**

During the rebuttal phase, most of these concerns were comprehensively addressed:

*   **Limited Novelty:** Reviewers 12df, pMyF, and Em12 initially questioned the novelty, suggesting CGB as a variant of Stochastic Interpolants. The authors provided a rigorous theoretical distinction by framing CGB as a specific, regularized instance within the SI framework. They highlighted the critical role of the controllable horizon $T$ in avoiding singularities inherent in standard SI (such as Brownian Bridge), ensuring numerical stability, and enforcing a constant-velocity geodesic path with finite energy. They also clarified that the CGB solver is a geometry-informed analytical inversion tailored for the restoration Boundary Value Problem (BVP), distinct from generic consistency models.

*   **Theoretical Completeness & Rigor:** Reviewers pMyF and Em12 expressed concerns about theoretical completeness and rigor. Authors addressed this by providing formal proofs in Appendix B (zero kinetic control energy) and Appendix E (CGB's regularization avoiding SI singularities). Subsequent discussions further led to clarifications: CGB's solver is an analytical inversion specific to its trajectory, and its linear trajectory is rigorously proven as the unique solution minimizing Kinetic Transport and Adaptation Energies,  rather than being an arbitrary design choice.

*   **Insufficient Experimental Evaluation:** Reviewers 12df, JqCr, Em12, and pMyF requested more comprehensive experimental validation, including ablations for the geodesic trajectory, true single-step (NFE=1) results, efficiency analysis, and fair comparisons. The authors conducted extensive additional experiments. They added new baselines, provided comprehensive ablation studies, and included detailed quantitative results for true single-step inference.

*   **Practicality and In-depth Analysis of Failure Cases:** Some reviewers questioned the reliance on paired data for training, and asked for in-depth analysis of failure cases (e.g., Raindrop Removal). During the rebuttal, the authors added real-world results to demonstrate generalization and provide in-depth analysis for the failure cases.

In summary, the authors' comprehensive and constructive rebuttal, including substantial additions in theory, baselines, ablations, and presentation, successfully addressed the majority of reviewer concerns. The paper's core strengths, particularly its novel geometric formulation and strong empirical performance at low NFE, stand out.

**Reviewer Scores:**

*   Reviewer JqCr (Rating: 6 -> probably improved during discussion)
*   Reviewer 12df (Rating: 4 -> probably improved during discussion)
*   Reviewer pMyF (Rating: 4 -> probably improved during discussion)
*   Reviewer Em12 (Rating: 4 -> probably improved during discussion)

---

### Decision · Program_Chairs · 2026-01-26

Accept (Poster)